

# Snow scavenging and phase partitioning of nitrated and oxygenated aromatic hydrocarbons in polluted and remote environments in central Europe and the European Arctic

Pourya Shahpoury[1,2], Zoran Kitanovski[1,3], Gerhard Lammel[1,4]

[1]Multiphase Chemistry Department, Max Planck Institute for Chemistry, Mainz, Germany

[2]Air Quality Research Division, Environment and Climate Change Canada, Toronto, Canada

[3]Department for Food Chemistry, National Institute of Chemistry, Ljubljana, Slovenia

[4]Research Centre for Toxic Compounds in the Environment, Masaryk University, Brno, Czech Republic

*Correspondence to*: Pourya Shahpoury (p.shahpoury@mpic.de)

**Abstract**

Nitrated and oxygenated polycyclic aromatic hydrocarbons (N/OPAHs) are emitted in combustion processes and formed in polluted air. Their precipitation cycling has hardly been studied. Fresh snow samples at urban and rural sites in central Europe, as well as surface snow from a remote site in Svalbard were analysed for 17 NPAHs, 9 OPAHs, and 11 nitrated mono-aromatic hydrocarbons (NMAHs), of which most N/OPAHs as well as nitrocatechols, nitrosalicylic acids, and 4-nitroguaiacol are studied for the first time in precipitation. In order to better understand the scavenging mechanisms, the particulate mass fractions ($\Theta$) at 273K were predicted using a multiphase gas-particle partitioning model based on polyparameter linear free energy relationships. ∑NPAH concentrations were 1.2-17.6 and 8.8-19.1 ng L$^{-1}$ at urban and rural sites, whereas ∑OPAHs were 0.3-1.1 and 0.5-2.4 µg L$^{-1}$ at these sites, respectively. Acenaphthoquinone and 9,10-anthraquinone were predominant in snow dissolved and particulate phase, respectively. NPAHs were only found in the particulate phase with 9-nitroanthracene being predominant followed by 2-nitrofluoranthene. Among NMAHs, 4-nitrophenol showed the highest abundance in both phases. The levels found for nitrophenols were in the same range or lower than those reported in the 1980s and 1990s. The lowest levels of ∑OPAHs and ∑NMAHs were found at the remote site (9.2 and 390.5 ng L$^{-1}$, respectively). N/OPAHs preferentially partitioned in snow particulate phase in accordance with predicted $\Theta$, whereas NMAHs were predominant in the dissolved phase, regardless of $\Theta$. It is concluded that the phase distribution of non-polar N/OPAHs in snow is determined by their gas-particle partitioning prior to snow scavenging, whereas that for polar particulate phase substances, i.e. NMAHs, is determined by an interplay between gas-particle partitioning in the aerosol, particle mass size distribution, and dissolution during in- or below-cloud scavenging.



# 1 Introduction

Nitrated and oxygenated polycyclic aromatic hydrocarbons (N/OPAHs) are formed primarily by oxidation of parent PAHs during combustion of fossil fuels as well as biomass burning, and secondarily through reactions of PAHs with atmospheric oxidants, such as $O_3$, OH and $NO_x$ (Walgraeve et al., 2010; Bandowe and Meusel, 2017). N/OPAHs were found in emissions from gasoline, diesel and biodiesel (Pham et al., 2013; Zielinska et al., 2004; Karavalakis et al., 2010), biomass and coal burning (Ding et al., 2012; Shen et al., 2012; Vicente et al., 2016; Shen et al., 2013a; Shen et al., 2013b; Huang et al., 2014) and solid waste incineration (Watanabe and Noma, 2009). These substance groups were also suggested to play a role in light absorption properties of biomass burning particulate matter (PM) (Lin et al., 2016). Some NPAHs, e.g. 3-nitrofluoranthene (3-NFLT) and 1-nitropyrene (1-NPYR), are associated specifically with combustion sources, whereas others such as 2-nitrofluoranthene (2-NFLT) and 2-nitropyrene (2-NPYR) are produced through gas phase reaction of FLT and PYR with OH radicals and NOx (Arey et al., 1986; Bandowe and Meusel, 2017). NPAHs are also formed through reactions in the aerosol condensed phase (Keyte et al., 2013; Jariyasopit et al., 2014). Photolysis of NPAHs results in the formation of other oxygenated and nitrated species such as hydroxynitro-PAHs, quinones, and nitrated quinones (Bandowe and Meusel, 2017). Unlike NPAHs, there is no agreement on distinct formation pathways of individual OPAHs – to various extents, they originate from both primary and secondary sources (Walgraeve et al., 2010; Zhuo et al., 2017). Many N/OPAHs are suggested to be more mutagenic than their parent species and are also classified as possible carcinogens (Finlayson-Pitts and Pitts, 2000; Lammel, 2015). Moreover, quinones, a prominent sub-class of OPAHs, are precursors of environmentally persistent free radicals (Borrowman et al., 2016) and reactive oxygen species (Chung et al., 2006; Charrier et al., 2014), inducing oxidative stress and inflammatory reactions in organisms, which may lead to cellular damage, respiratory and cardiovascular disease (Lodovici and Bigagli, 2011; Møller et al., 2014; Kelly and Fussell, 2017).

Nitrated mono-aromatic hydrocarbons (NMAHs) are composed of several chemically related compound classes such as: nitrophenols, nitroguaiacols, nitrocatechols and hydroxy-nitrobenzoic (i.e. nitrosalicylic) acids, which among others contain nitro, hydroxyl and carboxyl functionalities (Chow et al., 2016). Nitrophenols are emitted from primary sources, particularly biomass burning but also traffic exhaust, or formed in secondary processes – nitration of precursors such as phenol in the atmosphere (Harrison et al., 2005). Nitrocatechols and nitrosalicylic acids are mainly secondary oxidation products of substituted phenols (alkylphenols, methoxyphenols, hydroxybenzoic acids) emitted in primary aerosols from biomass (e.g. wood) burning (Iinuma et al., 2007; Iinuma et al., 2010; Kitanovski et al., 2012; Kahnt et al., 2013; Chow et al., 2016). NMAHs are ecotoxic (Pflieger and Kroflič, 2017) while little is known about human toxicity (Huang et al., 1995; Harrison et al., 2005; Kovacic and Somanathan, 2014). Due to nitrophenol phytotoxic potential, the research on them in precipitation was fostered during the late 1980s and early 1990s in relation with research on forest decline in central Europe (Rippen et al., 1987; Leuenberger et al., 1988; Herterich and Herrmann, 1990). Last but not least, NMAHs significantly contribute to the light absorptive properties of PM organic carbon (Mohr et al., 2013; Hinrichs et al., 2016; Bluvshtein et al., 2017; Teich et al., 2017) and can influence climate by altering Earth's albedo.

Semi-volatile organic compounds (SOCs) in the atmosphere are subject to removal by dry particle deposition and wet scavenging. The latter consists of two processes – i.e. particle scavenging and gas scavenging. The particle scavenging



is relevant for SOCs that show higher affinity towards particulate phase (Shahpoury et al., 2015). This is affected by SOC gas-particle partitioning (GPP) in the aerosol, a process controlled mainly by the substance molecular structure, PM chemical composition, and ambient temperature (Shahpoury et al., 2016). The magnitude of SOC sorption to PM is defined by its particulate mass fraction, $\Theta$ Eq. (1):

$$\Theta = c_p /(c_p + c_g) \tag{1}$$

where $c_p$ and $c_g$ are concentrations of SOC (ng m$^{-3}$) in the particulate and gas phase, respectively. The gas scavenging is relevant for substances which demonstrate minimum interaction with PM and therefore remain to various extents in the gas phase. Although gas scavenging is affected by GPP, the actual removal from the atmosphere is due to substance dissolution in cloud or rain droplets or sorption to snowflakes or other ice hydrometeors (Hoff et al., 1995; Bartels-Rausch et al., 2014). Following wet scavenging, SOC in the gas and particulate phases in the atmosphere

accumulate in precipitation dissolved and particulate phases, respectively. The fraction of SOC removed by particle scavenging is given by $\Theta_w$ Eq. (2):

$$\Theta_w = c_{pp} /(c_{pp} + c_{pd}) \tag{2}$$

where $c_{pp}$ and $c_{pd}$ are analyte concentrations (ng L$^{-1}$) in precipitation particulate and dissolved phases, respectively. It has been shown that the magnitude of particle scavenging is generally higher than gas scavenging for *hydrophobic* SOCs such as PAHs (Atlas and Giam, 1988; Bidleman, 1988; Shahpoury et al., 2015). In principle, for such substances

one could apply $\Theta$ as an indicator for predicting the SOC wet *particle* scavenging – i.e. the higher the $\Theta$ at a given temperature, the more efficient is the scavenging. However, it is not known if this concept also applies to *hydrophilic* SOCs which may also demonstrate high $\Theta$, and whether it is GPP or another process such as dissolution in water or a combination of both that determines the substance wet scavenging pathways.

There is currently very limited information in the literature about the occurrence of N/OPAHs and NMAHs in

precipitation, except for small number of OPAHs (Kawamura and Kaplan, 1983) and nitrophenols: 4-nitrophenol (4-NP), several methyl-nitrophenol isomers as well as dinitrophenols (2,4-dinitrophenol (2,4-DNP) and 2-methyl-4,6-dinitrophenol (i.e. dinitro-*ortho*-cresol, DNOC) were the most frequently measured nitrophenols in precipitation in urban and rural Europe (Leuenberger et al., 1988; Alber et al., 1989; Herterich and Herrmann, 1990; Levsen et al., 1990; Levsen et al., 1991; Levsen et al., 1993; Schüssler and Nitschke, 2001; Bossi et al., 2002; Kohler and Heeb,

2003; Asman et al., 2005; Belloli et al., 2006; Jaber et al., 2007; Schummer et al., 2009), in North America (Ganranoo et al., 2010), and Antarctica (Vanni et al., 2001) (Table S1).

The aims of the present study were (1) to investigate the presence of N/OPAHs and NMAHs in the dissolved and particulate phases of fresh snow, (2) estimate the substance particulate mass fractions in the atmosphere using a multiphase GPP model, based on poly-parameter linear free energy relationships (ppLFER), and (3) determine the

substance fractions removed by wet particle scavenging, and explore the effect of GPP vs. water solubility on target compound wet scavenging.



## 2 Experimental

### 2.1 Sampling

Snow samples were collected between winter 2015 and 2017 from three locations in Germany, i.e. Mainz (Ub1 and 4; urban-residential, ≈200000 inhabitants), Winterberg (Rr1) and Altenberg (Rr2; rural, >10 km from small towns),

two locations in Inn Valley, Austria, i.e. Götzens (Ub2; urban-residential of a mid-sized city, Innsbruck, ≈140000 inhabitants) and Kolsassberg (Rr3; rural, 10-20 km from city and towns), two locations in the Czech Republic, i.e. Ostrava (Ub3; urban, conurbation with ≈700000 inhabitants) and Pustá Polom (Rr4 and 5; rural, 20 km upwind from Ostrava), and one location in the Arctic, Tempelfjorden, Svalbard (Rm 1, remote, 50-80 km from small towns). The sample site details are shown in Table 1. Fresh snow samples (all sites, except Tempelfjorden) were collected by

placing several polypropylene trays (0.25 $m^2$ each) on the ground prior to snowfall. To this end, the snow forecast for a number of pre-selected sites was monitored on daily basis. Both collection trays and bottles were pre-cleaned prior to sample collection in the lab using detergent, tap water, deionized water, and high-purity ethanol. The snow was transferred with compaction in amber 2-L bottles and kept frozen at -18°C until analysis. Dry and light surface snow (0-5 cm deep, somewhat harder at the surface), which had fallen 3-2 days before, was collected at Tempelfjorden and

stored in pre-cleaned amber bottles. Following this sample collection and storage procedure, we rely on exclusion of significant phase change during storage and prior to analysis.

### 2.2 Sample processing

The samples were thawed at room temperature in the lab, and the meltwater was passed through a pre-assembled filtration-extraction setup (Fig. S1), allowing simultaneous separation of meltwater particulate phase and extraction

of dissolved phase. The setup consisted of a pre-assembled sterile analytical filter funnel (250 mL, Nalgene, Thermo Scientific, Waltham, MA, USA), connected to a solid-phase extraction disk (Bakerbond Speedisk, J.T. Baker, the Netherlands) using a Teflon adaptor designed in-house (Fig. S1). This was assembled on a J.T. Baker extraction station connected to a vacuum pump. A steady sample flow was established between the filter funnel and Speedisk throughout sample processing, by occasionally applying vacuum, where needed. The pH of meltwater samples was 4.5-5.

### 2.3 N/OPAH extraction and chemical analysis

0.22 µm cellulose nitrate filter in 250 mL analytical filter funnel and octadecyl (C18) Speedisk were used for N/OPAHs extraction. Speedisks were pre-conditioned with 50 mL of methanol followed by 10 mL of deionized water, and spiked with a mixture of deuterated standards, containing 75 ng of 1-nitronaphthalene-d₇, 2-nitrofluorene-d₉, 9-nitroanthracene-d₉, 3-nitrofluoranthene-d₉, 1-nitropyrene-d₉, 6-nitrochrysene-d₁₁, 9,10-anthraquinone-d₈, and 9-

fluorenone-d₈. Filter papers containing particulate phase were spiked with the same standard mixture after sample processing prior to their extraction. After loading the samples, Speedisks were capped with aluminium foil and dried by pumping air through them for 5 min.



The filter papers containing the particulate phase were extracted using a previous method (Albinet et al., 2014). Briefly, each filter paper was placed in a glass centrifuge tube (Duran, Schott, Mainz, Germany) and added with 7 mL of dichloromethane (DCM). The centrifuge tubes were capped with screw caps containing PTFE lining. Each sample was vortexed for 1.5 min, passed through a glass funnel plugged with a small amount of deactivated glass wool (in

order to remove residual sample matrix), and concentrated to 0.5 mL using a Turbovap II (Biotage, Uppsala, Sweden). The extracts were later loaded on a pre-conditioned 500 mg $SiO_2$ cartridge (Macherey-Nagel, Weilmünster, Germany), and eluted with 9 mL of 65:35 $n$-hexane-DCM. The dissolved phase samples enriched on C18 Speedisks were eluted with 40 mL of 1:1 $n$-hexane-DCM. All extracts from particulate and dissolved phase were concentrated to 0.5 mL and the solvent was exchanged to ethyl acetate. The sample volumes were further adjusted to 0.3 mL and transferred to 2-

mL vials containing pre-baked 0.4-mL glass inserts for further analysis. All solvents used for N/OPAH analysis were Suprasolv grade (Merck, Darmstadt, Germany).

The samples were analysed using a Trace 1310 gas chromatograph (GC, Thermo Scientific, Waltham, MA, USA) coupled to a TSQ8000 Evo triple-quadrupole mass selective detector (MS/MS, Thermo Scientific) in negative chemical ionization and selected ion monitoring (SIM) modes. The analyte separation was achieved on a J&W DB-

5ms column (30m + 10 m integrated guard, 0.25 mm ID, 0.25 μm film thickness, Santa Clara, CA, USA) with helium (99.9999%; Westfalen AG, Münster, Germany) as carrier gas at 1 mL $min^{-1}$ flow rate. The GC operating conditions were as follows: the GC oven was held at 60°C for 2 min, then ramped to 180°C at 15 °C $min^{-1}$, followed by a 5°C $min^{-1}$ ramp to 280°C and final hold time of 15 min. The injection port temperature was set to 250°C and operated in pulsed splitless mode (30 psi pulsed pressure for 1.5 min, and splitless time of 1.8 min). MS transfer line and ion

source temperature were set to 290 and 230°C, respectively. Methane (>99.9995, Messer, Bad Soden, Germany) was used as ionization gas with 1.5 mL $min^{-1}$ flow rate. Emission current and electron energy were set to 100 μA and -70 eV, respectively. The samples were analysed for N/OPAHs listed in Table 2. Each target analyte was identified using its retention time and quantification ion (Table 2). The analyte quantification was done using the internal method with 11-point calibration curves ranging from 0.25-1000 pg $μL^{-1}$.

**2.4 NMAH extraction and chemical analysis**

0.22-micron cellulose acetate filter in the 250 mL analytical filter funnel and divinylbenzene hydrophilic Speedisk were used for NMAH extraction. The detailed analytical method is described in a companion paper (Kitanovski and Naumoska, in preparation). Briefly, the pre-conditioned Speedisk was spiked with 100 ng of 4-nitrophenol-$d_4$, the sample (250 mL) was acidified with 2 mL of formic acid, and passed through the disk. The elution of NMAHs from

the disks was done using mixture of acetonitrile-methanol containing 3.4 μM ethylenediaminetetraacetic acid (EDTA). The presence of EDTA in the elution solvent was necessary for complete recovery of the NMAHs from the polymeric disks. The SPE extracts were further concentrated to near dryness using Turbovap II and later dissolved in 3:7 methanol-ammonium formate buffer (pH 3) containing EDTA. The PM retained on the cellulose acetate filters was spiked with the same quantity of 4-nitrophenol-$d_4$ and extracted using a previously published procedure (Kitanovski

et al., 2012). Briefly, the particles were extracted using methanol containing 3.4 μM EDTA with agitation in an



ultrasonic bath. The final extracts were concentrated to near dryness, and dissolved in 3:7 methanol-ammonium formate buffer (pH 3) containing EDTA. All samples were analyzed using a 1200 Series liquid chromatograph (LC; Agilent Technologies, Santa Clara, CA, USA) coupled to a 6130 single-quadrupole MS (Agilent Technologies) with an electrospray ionization (ESI) source. Separation was achieved on an Atlantis T3 column (150 × 2.1 mm ID, 3 μm;

Waters, Milford, MA, USA), thermostated at 30°C during sample analysis. The NMAH elution was done using 30:15:55 methanol-tetrahydrofuran-aqueous ammonium formate buffer (5 mM, pH 3) mobile phase in isocratic mode. The deprotonated NMAHs [M-H]⁻ listed in Table 2 were detected in negative ion ESI and SIM modes. The analyte quantification was done using the internal calibration method in concentration range 1-500 pg μL⁻¹.

### 2.5 Quality control

Field blanks were prepared during sample collection by exposing the pre-cleaned sample bottles with open cap to air for 5 min at the sites. The inner wall of the bottles was rinsed with 500 mL of deionized water in the lab and processed as field blank along with the rest of samples. Limits of quantification (LOQ) for analytes were calculated based on instrument detection limits (IDL), which in turn are determined using 3 times the chromatogram baseline noise level. IDL values were used in cases where analyte concentrations in blanks were <IDL. Where analyte concentrations in

samples exceeded the limit of quantification (LOQ: mean blank concentrations +3 standard deviations), the mean blank concentrations were subtracted from those in the corresponding samples.

### 2.6 Estimation of particulate mass fractions

The $\Theta$ for target analytes were estimated using modelled GPP constants, $K_P$ (m³g⁻¹ at 273.15 K), which were calculated using a multiphase ppLFER model (Shahpoury et al., 2016). The model differentiates between various organic and

inorganic phases of PM, and accounts for absorption into water soluble organic matter (WSOM) and organic polymers (OP), as well as adsorption onto black carbon, (NH₄)₂SO₄ and NaCl, Eq. (3):

$$K_P \ (\mathrm{m_{air}^3 \ g_{PM}^{-1}}) = \left[ \left( K_{BC} \times a_{BC} \times f_{BC} + K_{(NH_4)_2SO_4} \times a_{(NH_4)_2SO_4} \times f_{(NH_4)_2SO_4} + K_{NaCl} \times a_{NaCl} \right. \right. \\ \left. \left. \times f_{NaCl} \right) + \left( K_{DMSO} / \rho_{DMSO} \times f_{WSOM} + K_{PU} \times f_{OP} \right) \right] \tag{3}$$

where $K_{BC}$, $K_{(NH_4)_2SO_4}$, and $K_{NaCl}$ are the target substance partitioning coefficients (mol m⁻²surface/mol m⁻³air) for black carbon/diesel soot, (NH₄)₂SO₄ and NaCl (the last two represent secondary inorganic aerosols), respectively, $a_{BC}$, $a_{(NH_4)_2SO_4}$, and $a_{NaCl}$ are the adsorbent specific surface areas (m²surface g⁻¹adsorbent), and $f_{BC}$, $f_{(NH_4)_2SO_4}$, and $f_{NaCl}$ are their

mass mixing ratios in PM (gadsorbent g⁻¹PM). For $a_{BC}$, the geometric mean of 18.21 m² g⁻¹ was calculated from the values reported for traffic, wood, coal, and diesel soot (Jonker and Koelmans, 2002), whereas, $a_{(NH_4)_2SO_4}$, and $a_{NaCl}$ of 0.13 and 0.10 m² g⁻¹ were taken from Goss et al., (2003). $K_{DMSO}$ (m³air m⁻³DMSO) and $K_{PU}$ (m³air g⁻¹PU) are the substance partitioning (absorption) coefficients for dimethyl sulfoxide-air and polyurethane-air partitioning systems; $\rho_{DMSO}$ is dimethyl sulfoxide density (g m⁻³); $f_{WSOM}$ and $f_{OP}$, are mass mixing ratios of absorbing phases (gadsorbent g⁻¹PM),



corresponding to $f_{OM}$ (the mixing ratio of total organic matter in PM) $\times$ 0.60 and $f_{OM} \times 0.40$, respectively. The correction factors of 0.60 and 0.40 were estimated based on the data from Rogge et al., (1993). We assumed two scenarios for model calculations: $f_{BC} = 0.03$ and $f_{OM} = 0.30$, and $f_{BC} = 0.06$ and $f_{OM} = 0.60$. This resulted in $f_{WSOM}$ and $f_{OP}$ of 0.18 and 0.12, and 0.36 and 0.24 for the two scenarios, respectively. The contribution of inorganic salts to the overall

sorption process was neglected. The individual partitioning constants used in the multi-phase model are calculated using substance-specific Abraham descriptors listed in Table S2 and ppLFER models listed in Table S3 (Abraham et al., 2010; Kamprad and Goss, 2007; Roth et al., 2005; Goss et al., 2003). See Shahpoury et al., (2016) for more details about calculation with multiphase model and Endo and Goss (2014) for background about ppLFER concept. The predicted $K_P$ values were converted to $\Theta$ under two scenarios with $c_{PM}$ of 25 and 50 µg m$^{-3}$, Eq. (4):

$$\theta = \frac{K_P\, c_{PM}}{(1 + K_P\, c_{PM})} \qquad (4)$$

One must note that the ppLFER model used here predicts $K_P$ of a substance in neutral form. In particulate phase, depending on pH of the PM aqueous phase and p$K_a$ of the target substance, NMAHs may partly or completely deprotonate, resulting in enhanced solubility of the substance in the aqueous phase (Ahrens et al., 2012). Given the pH of samples in our study (i.e. 4.5-5), we expect 5-nitrosalicylic acid (5-NSA; see Table 2 for compound abbreviations), p$K_a$: 1.95 at 298 K (Aydin et al., 1997) and 3-nitrosalicylic acid (3-NSA; we expect similar p$K_a$ as that

of 5-NSA) to be completely deprotonated in PM aqueous phase, whereas 2,4-DNP and DNOC, p$K_a$: 4.07 at 298 K (Lide, 2010) and 4.48 at 293 K (WHO, 2000), respectively, will be partly dissociated. This implies that the actual partitioning could be under-predicted. The rest of NMAHs would be present in protonated form in our study (p$K_a$: 4-NP: 7.15, 2-methyl-4-nitrophenol (2-M-4-NP): 7.33, 3-methyl-4-nitrophenol (3-M-4-NP): 7.33, 4-nitrocatechol (4-NC): 6.93 at 298 K; we expect p$K_a$ values for 4-methyl-5-nitrocatechol (4-M-5-NC) and 3-methyl-5-nitrocatechol (3-

M-5-NC) to be close to that for 4-NC).

### 2.7 Air mass history analysis

The HYSPLIT (Draxler and Rolph, 2003) model was used to identify air mass histories related to the snowfall events over 3 days. The meteorological data (1°x1° resolution) used were from the Global Data Assimilation System (GDAS, NCEP, USA). Air mass changes were identified based on weather charts (Berliner Wetterkarte, 2015), except for

sample site in Svalbard, Rm1, as the snow fell 2-3 days prior to sample collection.

## 3 Results and discussion

### 3.1 Air mass backward trajectory analysis

For all central European sites, the air masses corresponding to the snow samples had been advected mostly from westerly directions (see air mass trajectories in Fig. S2), passing over polluted areas of central and western Europe

(such as in NE France/SW Germany, W and SE Germany for samples Ub1, Ub3, Rr1 and Rr2) until the last 100-200 km before precipitation started, when they had been transported over rural areas. The snowfalls leading to samples



Ub1, Ub3, Rr1, and Rr5 followed immediately frontal passages with advection from westerly directions (Fig. S2), unlike in the other precipitation events.

**3.2 N/OPAH concentrations and distribution in snow**

Snow dissolved and particulate phases were analysed for N/OPAHs following the method described in Sect. 2.3.

Among these compounds, NPAHs were only found in the snow particulate phase. On the contrary, OPAHs, 9-fluorenone (9-OFLN), acenaphthoquinone ($O_2ACE$), and 9,10-anthraquinone (9,10-$O_2ANT$) were found in nearly all dissolved phase samples, except at the remote site where 9,10-$O_2ANT$ was not found (Fig. 1A). $O_2ACE$ demonstrated the largest concentration range (4.1 to 779.8 ng $L^{-1}$) followed by 9,10-$O_2ANT$ (< LOQ in Rm 1 sample to 89.7 ng $L^{-1}$), and 9-OFLN (2.6 to 45.3 ng $L^{-1}$). 1,2-benzanthraquinone (1,2-$O_2BAA$) was not found in samples Ub1 and Rm1,

while concentrations ranged from 0.2 to 1.3 ng $L^{-1}$ for the rest of the samples (Fig. 1A). 1,4-naphthoquinone (1,4-$O_2NAP$), benzanthrone (OBAT), and benz(*a*)fluorenone (BaOFLN) were found less frequently with concentrations up to 7.1, 1.4, and 0.2 ng $L^{-1}$, respectively. Overall, the lowest OPAH concentrations were from the remote site (Rm1, $\sum$ OPAHs 7.3 ng $L^{-1}$), while sample Ub4 was the most polluted ($\sum$ OPAHs 834 ng $L^{-1}$), mainly due to the contribution of $O_2ACE$, followed by Rr1 (303.6 ng $L^{-1}$), and Rr3 (279.3 ng $L^{-1}$).

9-OFLN, 9,10-$O_2ANT$, 1,4-$O_2NAP$, and 1,2-$O_2BAA$ were previously found in diesel exhaust (Choudhury, 1982; Cho et al., 2004) and biomass and coal burning emission (Shen et al., 2013a; Huang et al., 2014; Vicente et al., 2016, along with $O_2ACE$), whereas in ambient PM, 9-OFLN 9,10-$O_2ANT$, and 1,2-$O_2BAA$ were attributed to both primary and secondary sources (Kojima et al., 2010; Souza et al., 2014; Lin et al., 2015; Zhuo et al., 2017). The contribution of primary sources is expected to be higher during the cold season with heating activities dominating the vehicular

emission (Lin et al., 2015). OBAT does not have a stable parent PAH precursor with the same number of rings in the atmosphere (Kojima et al., 2010) and, along with BaOFLN and benz(*b*)fluorenone (BbOFLN), was associated with primary combustion sources (Albinet et al., 2007; Karavalakis et al., 2010; Shen et al., 2013b; Souza et al., 2014; Huang et al., 2014; Tomaz et al., 2016; Vicente et al., 2016).

In the particulate phase of snow, four NPAHs and seven OPAHs were detected (Fig. 1B). In Rm1 only three OPAHs,

$O_2ACE$ (1.7 ng $L^{-1}$), BaOFLN (0.15 ng $L^{-1}$), and 1,2-$O_2BAA$ (0.13 ng $L^{-1}$), were found with relatively low concentrations. Among all detected analytes, 9,10-$O_2ANT$ (found in Ub2, Rr1-3) showed the highest concentrations ranging from 155.6 to 242.2 ng $L^{-1}$ followed by 9-OFLN (found in all but Rm 1, 4.0-30.3 ng $L^{-1}$), BaOFLN (0.15-27.3 ng $L^{-1}$), and 1,2-$O_2BAA$ (0.13-23.3 ng $L^{-1}$). The detected NPAHs, 9-nitroanthracene (9-NANT), 2-nitrofluoranthene (2-NFLT), 1-nitronaphthalene (1-NNAP), and 2-nitronaphthalene (2-NNAP) showed relatively low

concentrations ranging from <LOQ-13.6 ng $L^{-1}$ (9-NANT), <LOQ-2.6 ng $L^{-1}$ (2-NFLT), <LOQ-1.3 ng $L^{-1}$ (1-NNAP), and <LOQ - 0.32 ng $L^{-1}$ (2-NNAP), respectively. NPAHs found in the present study are the most frequently detected congeners in the gas (1- and 2-NNAP) and particulate (2-NFLT and 9-NANT) phases (Dimashki et al., 2000; Bamford and Baker, 2003; Albinet et al., 2006; Tomaz et al., 2016; Bandowe and Meusel, 2017), with 2-NFLT being exclusively formed through reaction of FLT with oxidants in the atmosphere (Bandowe and Meusel, 2017) and the

other three NPAHs being produced by both primary and secondary sources (Zhuo et al., 2017). One must also note



the possibility of NPAH conversion to OPAHs in the atmosphere. For instance, formation of 1,4-$O_2$NAP and 9,10-$O_2$ANT following photolysis of 1-NNAP and 9-NANT was suggested by previous studies (Keyte et al., 2013; Bandowe and Meusel, 2017); this might have contributed to the observed concentration patterns in the present study. Overall, the lowest and highest $\sum$ N/OPAH concentrations in the particulate phase were found at Rm1 (1.9 ng $L^{-1}$) and

Rr3 (359.8 ng $L^{-1}$), with NPAHs contributing up to 10% to the total concentrations across the samples.

Snow samples Ub3 and Rr4-5 were not phase separated for N/OPAH analysis. Of all targeted NPAHs at these sites, only 2-NFLT (11.6-19.1 ng $l^{-1}$) was found in the samples (Fig. 1C). Among OPAHs, OBAT, 9,10-$O_2$ANT, and $O_2$ACE showed the highest concentrations ranging from 478.3 - 758.1, 234.9 - 607.7, and 141.8 - 609.9 ng $L^{-1}$, respectively, whereas 1,2-$O_2$BAA, BbOFLN, BaOFLN, 9-OFLN, and 1,4-$O_2$NAP demonstrated relatively lower levels, up to 136.5,

106.5, 95.9, 57.6, and 18.6 ng $L^{-1}$, respectively (Fig. 1C). OPAH concentrations were overall higher in Rr4-5 samples than Ub3. The predominance of OBAT, 9,10-$O_2$ANT, $O_2$ACE, and 9-OFLN is consistent with findings of the previous studies for ambient PM and primary emission samples. High abundance of OBAT is of particular concern because this compound is precursor of the mutagenic 2- and 3-nitrobenzanthrone (Enya et al., 1997; Phousongphouang and Arey, 2003).

**3.3 NMAH concentrations and distribution in snow**

NMAHs targeted for analysis were found in all dissolved phase samples, with the exception of 4-nitroguaiacol (4-NG; Fig. 2A-B). 4-NP showed the highest concentrations ranging from 9.5 to 2155.4 ng $L^{-1}$, followed by 4-NC (18.2- 763.6 ng $L^{-1}$), 3-M-4-NP (4.6 - 547.3 ng $L^{-1}$), 2-M-4-NP (18.4 - 341.1 ng $L^{-1}$), and 5-NSA (6.6 - 313.5 ng $L^{-1}$). In this phase, 4-NG was exclusively found in samples from urban (Ub3 and Ub4, 155.2 and 284.7 ng $L^{-1}$, respectively; Fig. 2B),

rural (Rr4 and Rr5, 169.4 and 282.3 ng $L^{-1}$, respectively; Fig. 2A) and remote (Rm1, 14.8 ng $L^{-1}$; Fig. 2A) environments. Overall, the dissolved phase samples from Rm1 and Ub3 were the least polluted ($\sum$ NMAHs: 387.0 and 746.7 ng $L^{-1}$) while Ub2, Rr1, and Rr4-5 showed the highest concentrations of target compounds (1231 - 1345 ng $L^{-1}$) across the samples (Fig. 2A-B).

The NMAHs were found more sporadically in the snow particulate phase. 4-NP was found in all samples, with

concentrations ranging from 2.2 (Rr3) to 106.9 (Ub4) ng $L^{-1}$ (Fig. 2C-D). In fact, 4-NP was the only NMAH found at the remote site Rm1 (3.5 ng $L^{-1}$; Fig. 2C). 4-NG was only found in Rr2 (18.5 ng $L^{-1}$; Fig. 2C) and Ub4 (40.8 ng $L^{-1}$; Fig. 2D) samples. 3- and 5-NSA were found in Ub4 particulate phase (2.9 and 13.5 ng $L^{-1}$, respectively; Fig. 2D), while 4- and 3-M-5-NC were found in Ub4 (5.8 and 11.4 ng $L^{-1}$, respectively; Fig. 2D), Rr2 (4-M-5-NC: 0.8 ng $L^{-1}$; Fig. 2C) and Rr5 (3-M-5-NC: 1 ng $L^{-1}$; Fig. 2C). In addition, 2,4-DNP and 4-NC were only found in Ub4 (2.7 and

36.9 ng $L^{-1}$, respectively; Fig. 2D), Ub3 and Rr4 (up to 2.9 ng $L^{-1}$ for 2,4-DNP) and Rr1-2 (up to 1.3 ng $L^{-1}$ for 4-NC). The highest concentrations of DNOC and 3-M-4-NP in the particulate phase were from the Czech sites (up to 11.8 ng $L^{-1}$ for DNOC and 21.8 ng $L^{-1}$ for 3-M-4-NP), with DNOC concentration being ~ 2 times higher at the urban Ub3 than at the rural and upwind site Rr4 (Fig. 2C-D). The highest particulate concentrations for 2-M-4-NP were found in Ub4 sample (39.2 ng $L^{-1}$; Fig. 2D). These analytes (DNOC, 2-M- and 3-M-4-NP) together with 4-NP were the most

abundant compounds in the particulate phase. Overall, in terms of the levels of target NMAHs in the particulate phase,



the remote site (Rm1) was the cleanest ($\sum$ NMAHs 3.5 ng L$^{-1}$) and the Czech sites, with the urban exceeding the rural (52.5 and 33.5 ng L$^{-1}$ in Ub3 and Rr5, respectively) were the most polluted sites in our study.

The nitrophenol concentrations measured are in the same range with those previously reported for snow samples collected in central Europe and Antarctica (Alber et al., 1989; Vanni et al., 2001; Table S1), but usually lower (up to

two orders of magnitude lower) than those reported in rainwater from central and northern Europe (Leuenberger et al., 1988; Herterich and Herrmann, 1990; Levsen et al., 1991; Bossi et al., 2002; Asman et al., 2005; Schummer et al., 2009; Table S1) and the USA (Ganranoo et al., 2010; Table S1). One must note that here we report the NMAH concentrations in both particulate and dissolved phases, which was not done in the previous studies (Table S1). The dissolved phase concentrations of 4-NG in samples from urban and rural environments were comparable (~ 155-285

ng L$^{-1}$; Fig. 2A-B), indicating polluted and aged air masses with biomass burning origin (Kitanovski et al., 2014; Kroflič et al., 2015; Yang et al., 2016). These values are one order of magnitude higher than the measured concentration for the Arctic sample Rm1 (Fig. 2A). In urban samples with the exception of Ub3, nitrocatechols (4-NC and isomeric methyl-nitrocatechols) were the second most abundant NMAH species, following nitrophenols (Table S4A). This is in agreement with our current unpublished data and previous studies on winter PM samples

(Kitanovski et al., 2012). In contrast, in rural samples Rr3-4 and the remote sample Rm1, nitrosalicylic acids are the second most abundant NMAH species (Table S4A), which might indicate either higher stability or higher formation of these compounds during the transport of polluted air to the rural environments and the Arctic. Increased production of nitrocatechols in the urban environment due to anthropogenic activities, such as wood burning in winter, was suggested by previous studies (Kitanovski et al., 2012; Chow et al., 2016). Hydroxybenzoic acids, which are

precursors of nitrosalicylic acids, are primarily emitted during wood burning (Iinuma et al., 2007). Kitanovski et al., (2012) reported a high correlation between concentrations of nitrocatechols and nitrosalicylic acids (R$^2$ >0.8) in urban PM. This suggests secondary formation of nitrosalicylic acids as well as biomass burning as the major emission source, similar to the previous reports on nitrocatechols (Iinuma et al., 2010; Kitanovski et al., 2012; Kahnt et al., 2013; Chow et al., 2016; Caumo et al., 2016).

**3.4 Modelled particulate mass fractions and actual fractions removed by particle scavenging**

The target compound $\Theta$ was estimated following the method explained in Sect. 2.6. We examined three scenarios i.e., a lower, middle and upper scenario with regard to pollution: (1) with $f_{BC}$ and $f_{OM}$ of 0.03 and 0.30 and $c_{PM}$ of 25 µg m$^{-3}$, (2) $f_{BC}$ and $f_{OM}$ of 0.06 and 0.60 and $c_{PM}$ of 25 µg m$^{-3}$, and (3) $f_{BC}$ and $f_{OM}$ of 0.06 and 0.60 and $c_{PM}$ of 50 µg m$^{-3}$. In fact, the 24-h mean $c_{PM}$ for Mainz (Ub4) and the Ostrava (Ub3) sampling events was 25 and 27 µg m$^{-3}$, respectively.

No data are available from the other sites. As can be seen from Fig. 3A-C, $\Theta$ at 273 K is ~ 1 (complete sorption to PM) for 70% of the target compounds, independent of the above scenarios. The calculations showed that <20% of 1-NNAP, 2-NNAP, 1,4-O$_2$NAP, 9-OFLN, 2,4-DNP, and DNOC, and between 20 and 80% of 5-NACE, 2-NFLN, O$_2$ACE, 3-NSA, and 4-NG can be expected to be in the particulate phase at 273 K (Fig. 3). For these substances, $\Theta$ increases by up to two fold when increasing $f_{BC}$, $f_{OM}$, or $c_{PM}$. An interesting trend was seen for 2,4-DNP and 4-NP: the

presence of two nitro groups on 2,4-DNP is expected to promote stronger interactions with PM due to increased e-





donor ability, compared to 4-NP (one nitro group). However, our model calculations suggest the opposite pattern, i.e. complete sorption of 4-NP to PM, but <5% sorption for 2,4-DNP. This is largely related to intramolecular H-bonding between the H-atom of the hydroxy group and the O-atom of the nitro group in ortho position. This reduces the H-bonding ability of 2,4-DNP compared to 4-NP.

For substances which indicate complete sorption to PM, particle scavenging is expected to be the dominant source of wet deposition, whereas, for the rest of substances, which distribute between gas and particulate phases, both gas and particle scavenging are relevant. Accordingly, one would expect to find the substances with complete sorption to PM in precipitation particulate phase. Our observations depicted in Fig. 4A show that N/OPAHs in the present study were mainly associated with precipitation particulate phase. The exceptions were $O_2ACE$ (70-100%), 9-OFLN (30-100%),

and 9,10-$O_2ANT$ (10-100%) which were found to various extents in the dissolved phase. This trend is consistent with predicted values of $\Theta$, particularly for $O_2ACE$ and 9-OFLN, and to a lesser extent 9,10-$O_2ANT$ (Fig. 3B), and supports the implied assumption that phase partitioning in air is preserved in snow.

For one of the events, Ub3, the analyte concentrations in the gas and particulate phases of the air have been determined (24h sample collected until shortly after snowfall started; Table S4B) in addition to concentrations in precipitation.

The total scavenging ratios ($W_T$; unitless) calculated for the target N/OPAHs were $10^3$-$10^4$ (Table S4B), which fall within the range suggested for removal of polyaromatic compounds through particle scavenging (Shahpoury et al., 2015). With the exception of 1,4-$O_2NAP$ and 9-OFLN, the calculated ratios are consistent with modelled $\Theta$ of 0.81-1 (at 273 K) for the target N/OPAHs (Table S4B) in that particle scavenging was the dominant removal mechanism.

The observation for NMAHs, however, are considerably different (Fig. 4B). These substances were mainly found in

the dissolved phase, including the seven analytes for which complete sorption to PM was predicted under various scenarios. Among all, 4-NP showed the highest fraction in the precipitation particulate phase and, as predicted, higher than 2,4-DNP, followed by 2-M-4-NP. The observed opposite pattern may suggest that besides GPP another process influenced the wet scavenging of NMAHs and consequently their distribution in precipitation phases. We hypothesize that following both in- and below-cloud scavenging into sub-cooled droplets, particulate-phase NMAHs would

dissolve into the bulk water or water layer surrounding ice hydrometeors (Hoff et al., 1995). This process is affected by the substance water solubility and, therefore, is anticipated to be most efficient for highly water soluble substances. In fact, among N/OPAHs found in the present study, the only substance that showed high enrichment in the dissolved phase was $O_2ACE$ (Fig. 4A) with log octanol-water partitioning coefficient (log $K_{OW}$; unitless) of 1.95 compared to log $K_{OW}$ of 3.03-4.98 for the other detected N/OPAHs; Fig. S3). NMAHs are highly water soluble (s = 0.3-10 g $L^{-1}$,

log $K_{OW}$ = 0.64-2.54; Fig. S3) and, therefore, are expected to undergo the abovementioned process. This hypothesis is consistent with the results of a previous study where positive correlation was found between in-cloud scavenging efficiencies of PM-bound polar organics and the substance polarity and water solubility (Limbeck and Puxbaum, 2000). This process (which potentially results in predominance of NMAHs in the dissolved phase), however, must be distinguished from the gas scavenging which was proposed by Leuenberger et al., (1985) as a dominant process for

removal of volatile methylated phenols, where the predominance of these substances in the dissolved phase corresponded to their abundance in the gas phase.



In sample Ub3 which corresponds to known analyte concentrations in air particulate phase, $W_T$ was found to be in the range $10^3$-$10^5$ (Table S4B), which is relevant for wet *particle* scavenging of aromatic compounds (Shahpoury et al., 2015), and suggests that the removal of NMAHs followed a similar mechanism. Interestingly, the difference in $W_T$ between 3- and 5-NSA (~1.4 times higher for 5-NSA) closely resembled that of estimated $\Theta$ (~1.6 times higher for 5-NSA at 273 K) for these compounds (Table S4B). However, we found noticeable differences in $W_T$ between the NMAH subgroups, namely $W_T$ values were higher for nitrophenols ($1.3\times10^4$ – $1.6\times10^5$) and nitrosalicylic acids ($5.7\times10^4$ – $8.2\times10^4$) than nitrocatechols ($1.1\times10^3$ – $2.8\times10^3$; Table S4B). This pattern cannot be explained by the substance GPP (compare $\Theta$ in Fig. 3C), or the order of their water solubility. Accordingly, nitrosalicylic acids are the most water soluble substances among the three subgroups, regardless of their predicted $K_{OW}$ values shown in Fig. S3 (due to their pKa of < 2, these compounds would be completely deprotonated in our samples (pH 4-5), which enhances their water solubility. This is also reflected in them showing the lowest retention times in Table 2). Moreover, nitrocatechols are more water soluble (log $K_{OW}$ 0.64-1.27; Fig. S3) than nitrophenols (log $K_{OW}$ 1.68-2.43) – it must be noted that nitrocatechols and nitrophenols have pKa of > 6, which means they would be present in protonated (neutral) form in our samples. Given the above explanation, we believe that the observed differences in $W_T$ of the subgroups were in fact related to the mass size distribution of NMAHs in the atmosphere prior to snow scavenging. According to our unpublished data from Ub3, nitrophenols were mainly associated with coarse PM fraction (~60%), followed by nitrosalicylic acids (~35%). Nitrocatechols on the other hand were exclusively associated with the fine fraction (PM₁; ~80%), similar to observations reported by Li et al., (2016). This mass size distribution could potentially enhance snow scavenging of nitrophenols following the *interception* of coarse particles by snowflakes – this mechanism was suggested to be important for PM in the size range of 1 μm to a few microns (Zhang et al., 2013), and its contribution is expected to increase with the particle size. It was further suggested that a combination of Brownian diffusion, interception, and inertial impaction leads to a low *snow particle-aerosol particle collection efficiency* (suggested to be the most important source of uncertainty in snow scavenging of PM) in the particle size range of 0.01-1.0 μm (Zhang et al., 2013), which could further explain the relatively low $W_T$ values found for nitrocatechols in the present study. We conclude that an interplay between GPP and PM mass size distribution determined the mechanism and magnitude of NMAH wet scavenging, whereas water solubility contributed to their phase partitioning in snow.

**Final remarks**

The phase-separated concentrations of nitrated and oxygenated aromatic compounds were measured in snow from several locations in central Europe and European Arctic. For the first time, we have reported the snow concentrations for several classes of nitrated and oxygenated aromatics, such as nitrocatechols, nitrosalicylic acids, nitrated and oxygenated PAHs, and 4-nitroguaiacol. Our results showed that a model-predicted particulate mass fraction, $\Theta$, can be reliably applied in order to predict the scavenging mechanism (gas or particle scavenging) and efficiency of *hydrophobic* N/OPAHs and, hence, their distribution in precipitation phases i.e., dissolved and particulate. This suggests that the atmospheric lifetime of N/OPAHs in relation to removal processes follows the same pattern as that of parent PAHs (Sharma and McBean, 2002). On the contrary, $\Theta$ is not a good measure for predicting the wet removal



mechanism and distribution of *hydrophilic* NMAHs in precipitation. Our data suggests that the phase distribution of polar particulate phase substances, such as NMAHs in snow is determined by an interplay between GPP in the aerosol, mass size distribution in the atmosphere, and dissolution in cloud or rain droplets, or the liquid water phase on the surface of ice hydrometeors during in- or below-cloud scavenging. This behaviour is in line with what was previously

shown for the effect of aerosol mass size distribution on snow scavenging (Zhang et al., 2013), and what was found for in-cloud scavenging of other polar mono- and difunctional organics, such as aliphatic alcohols, and aliphatic and monoaromatic aldehydes and carboxylic acids (Limbeck and Puxbaum, 2000), namely the polarity of these substances is a significant parameter for their scavenging when solubility is $> 1$ g L$^{-1}$. The experimental data on GPP of NMAHs is scarce in the literature, which was also not addressed here. The contribution of gaseous polar NMAHs to the total

scavenging, beyond the scope of this study, should be addressed, similar to other polar aliphatic and aromatic organics (Limbeck and Puxbaum, 2000).

### Data availability

The dataset used in this manuscript is included in the Supplement, and further information is available from the corresponding author (p.shahpoury@mpic.de).

### Competing interests

The authors declare that they have no conflict of interest.

### Acknowledgements

We thank Paulo C. Alarcón, Thorsten Hoffmann (Max Planck Institute for Chemistry, Mainz), Pernilla Carlsson

(University Centre in Svalbard, Longyearbyen), Ulrike Nickus (University of Innsbruck), Libor Černikovský (Czech Hydrometeorological Institute, Ostrava), Roman Prokeš, Ondřej Šáňka, Petra Přibylová, and Petr Kukučka (RECETOX, Brno) for on-site and laboratory support, and Landesamt für Umwelt Rheinland-Pfalz (ZIMEN network) for PM data. We also thank Roland Kallenborn (Norwegian University of Life Sciences) for supporting us with his research infrastructure, and Michael H. Abraham (University College London) for providing descriptors for the

ppLFER model. This research was supported by the Max Planck Society and the Czech Science Foundation (#P503 16-11537S).

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





**Table 1. Sampling site details**

|  | Coordinates | Sampling date | Snowfall started | Sample collected |
|---|---|---|---|---|
| **Urban** | | | | |
| Ub1 Mainz | 49.99° N, 8.23° E | 23 Feb 2015 | 8:00 | 12:45 |
| Ub2 Götzens | 47.23° N, 11.31° E | 25 Feb 2015 | Overnight | 9:00 |
| Ub3 Ostrava | 49.86° N, 18.26° E | 19 Feb 2016 | 14:00 | 19:00 |
| Ub4 Mainz | 49.99° N, 8.23° E | 10 Jan 2017 | 9:00 | 15:00 |
| **Rural** | | | | |
| Rr1 Winterberg | 51.18° N, 8.49° E | 03 Mar 2015 | 15:00 | 18:30 |
| Rr2 Altenberg | 50.78° N, 13.69° E | 05 Mar 2015 | Overnight | 8:00 |
| Rr3 Kolsassberg | 47.28° N, 11.65° E | 25 Feb 2015 | Overnight | 10:00 |
| Rr4 Pustá Polom 1 | 49.86° N, 17.98° E | 19 Feb 2016 | 9:30 | 23:00 |
| Rr5 Pustá Polom 2 | 49.86° N, 17.98° E | 23 Feb 2016 | 14:00 | 23:00 |
| **Remote** | | | | |
| Rm1 Tempelfjorden | 78.45° N 17.32° E | 4 Mar 2015 | 1 Mar 2015 | after snowfall [a] |

[a] old snow, which had fallen 3-2 days before sampling





**Table 2. Target compound list**

| Analyte | Abbreviation | RT | Q1 |
|---|---|---|---|
| 1-Nitronaphthalene | 1-NNAP | 12.12 | 173.1 |
| 2-Nitronaphthalene | 2-NNAP | 12.62 | 173.1 |
| 5-Nitroacenaphthene | 5-NACE | 17.52 | 199.1 |
| 2-Nitrofluorene | 2-NFLN | 19.07 | 211.1 |
| 9-Nitroanthracene | 9-NANT | 19.46 | 223.1 |
| 9-Nitrophenanthrene | 9-NPHE | 20.64 | 223.1 |
| 3-Nitrophenanthrene | 3-NPHE | 21.4 | 223.1 |
| 2-Nitrofluoranthene | 2-NFLT | 25.75 | 247.1 |
| 3-Nitrofluoranthene | 3-NFLT | 25.80 | 247.1 |
| 1-Nitropyrene | 1-NPYR | 26.63 | 247.1 |
| 2-Nitropyrene | 2-NPYR | 26.95 | 247.1 |
| 7-Nitrobenz(a)anthracene | 7-NBAA | 29.41 | 273.1 |
| 6-Nitrochrysene | 6-NCHR | 30.66 | 273.1 |
| 1,3-Dinitropyrene | 1,3-N2PYR | 31.8 | 292.1 |
| 1,6-Dinitropyrene | 1,6-N2PYR | 32.81 | 292.1 |
| 1,8-Dinitropyrene | 1,8-N2PYR | 33.54 | 292.1 |
| 6-Nitrobenz(a)pyrene | 6-NBAP | 36.73 | 297.1 |
| 1,4-Naphthoquinone | $1,4\text{-}O_2NAP$ | 10.18 | 158.1 |
| 9-Fluorenone | 9-OFLN | 13.78 | 180.1 |
| 9,10-Anthraquinone | $9,10\text{-}O_2ANT$ | 17.03 | 208.1 |
| Acenaphthoquinone | $O_2ACE$ | 17.82 | 198.1 |
| 2-Nitro-9-fluorenone | 2-N-9-OFLN | 20.54 | 225.1 |
| Benz(a)fluorenone | BaOFLN | 22.88 | 230.1 |
| Benz(b)fluorenone | BbOFLN | 23.82 | 230.1 |
| Benzanthrone | OBAT | 25.07 | 230.1 |
| 1,2-Benzanthraquinone | $1,2\text{-}O_2BAA$ | 26.46 | 258.1 |
| 3-Nitrosalicylic acid | 3-NSA | 3.60 | 182 |
| 5-Nitrosalicylic acid | 5-NSA | 5.07 | 182 |
| 4-Nitrocatechol | 4-NC | 7.76 | 154 |
| 4-nitroguaiacol | 4-NG | 8.29 | 168 |
| 4-Methyl-5-nitrocatechol | 4-M-5-NC | 9.47 | 168 |
| 4-Nitrophenol | 4-NP | 10.00 | 138 |
| 2,4-Dinitrophenol | 2,4-DNP | 10.92 | 183 |
| 3-Methyl-4-nitrophenol | 3-M-4-NP | 13.19 | 152 |
| 3-Methyl-5-nitrocatechol | 3-M-5-NC | 14.01 | 168 |
| 2-Methyl-4-nitrophenol | 2-M-4-NP | 16.72 | 152 |
| Dinitro-ortho-cresol | DNOC | 17.05 | 197 |

Abbreviations, retention times (RT), and quantification ions (Q1) of target analytes




**Figure 1 N/OPAHs found in snow samples**

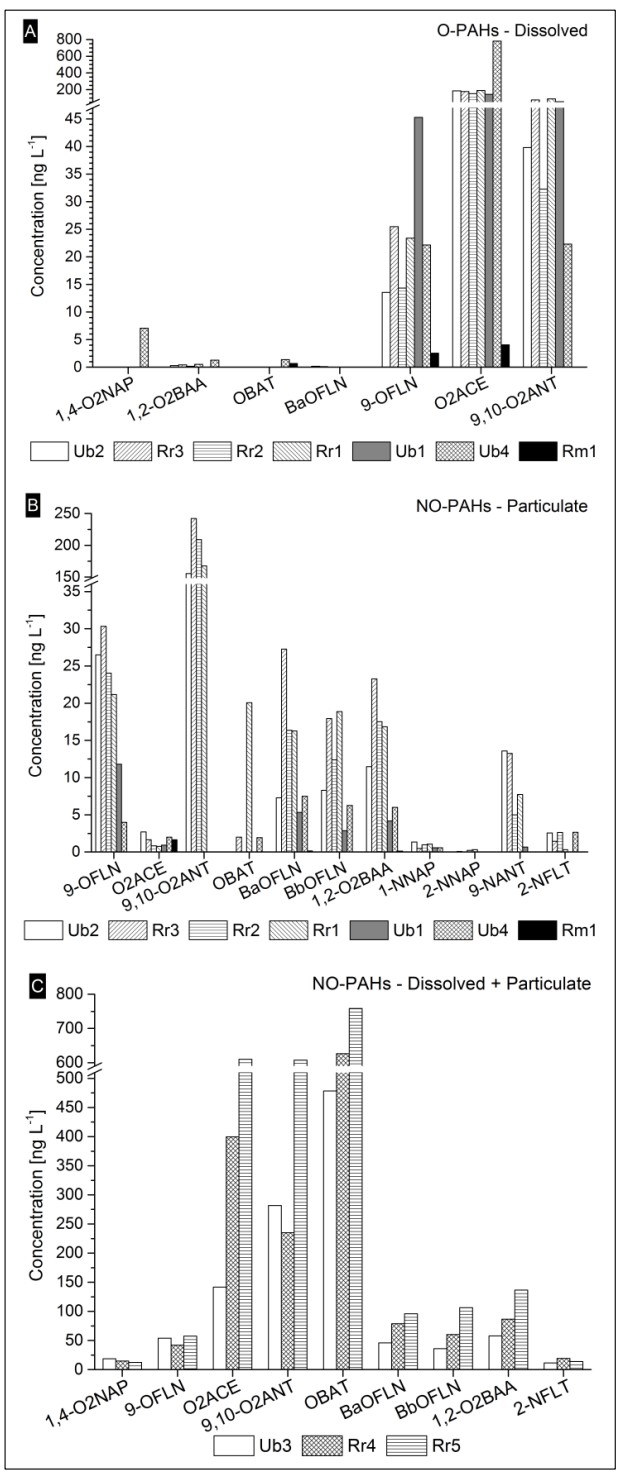



**Figure 2 NMAHs found in snow samples**

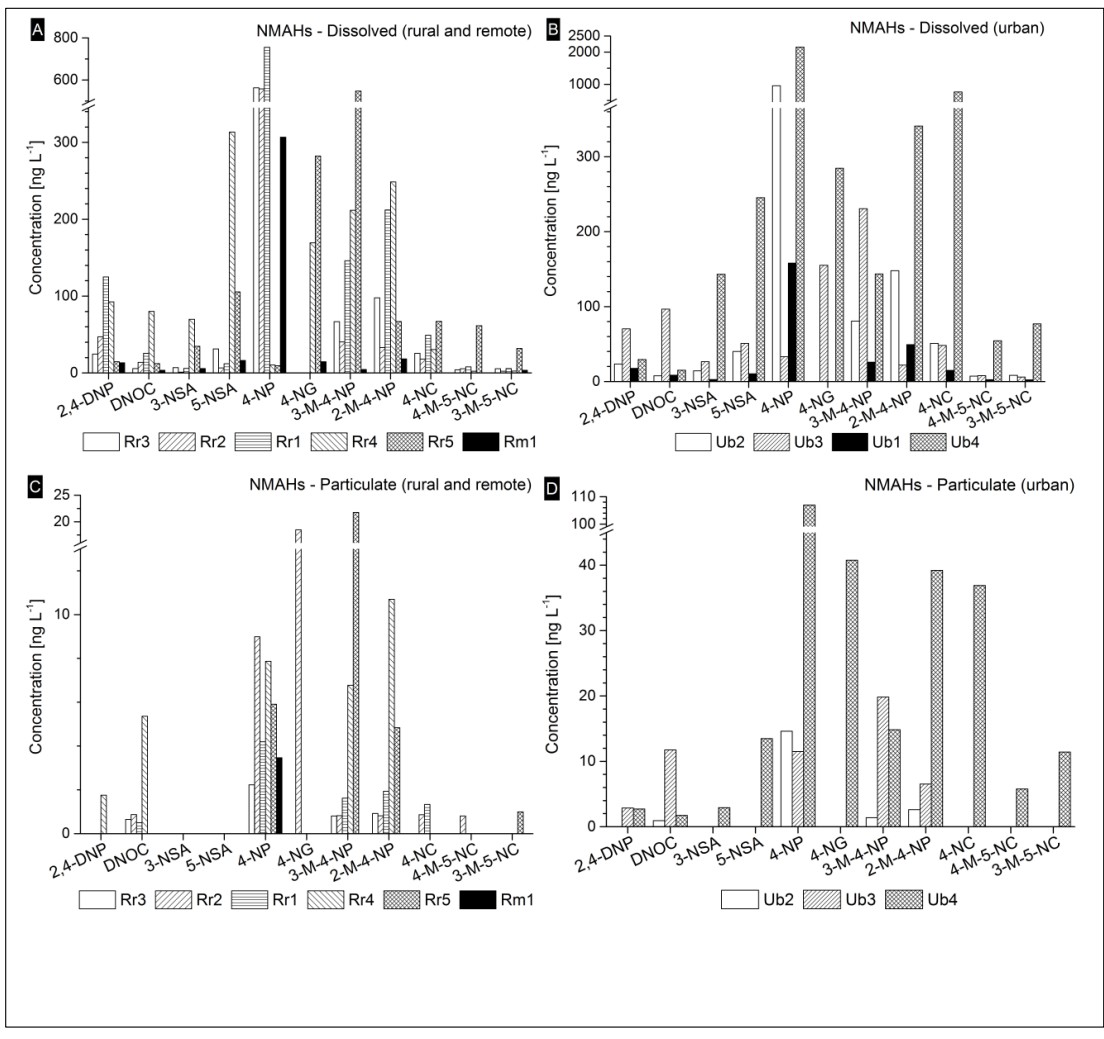





**Figure 3 Particulate mass fractions $\Theta$ estimated at 273 K using multiphase gas-particle partitioning model for NPAHs (A), OPAHs (B), and NMAHs (C)**

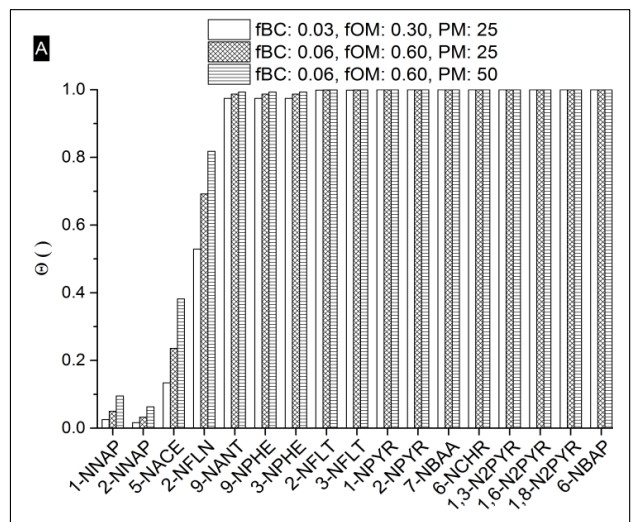

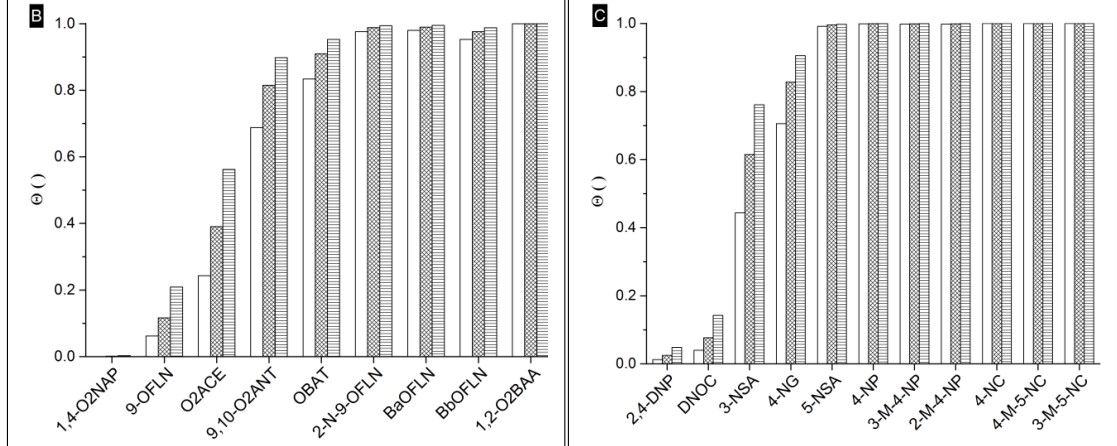



**Figure 4 Fraction of N/OPAHs (A) and NMAHs (B) removed by particle scavenging ($\Theta_W$)**

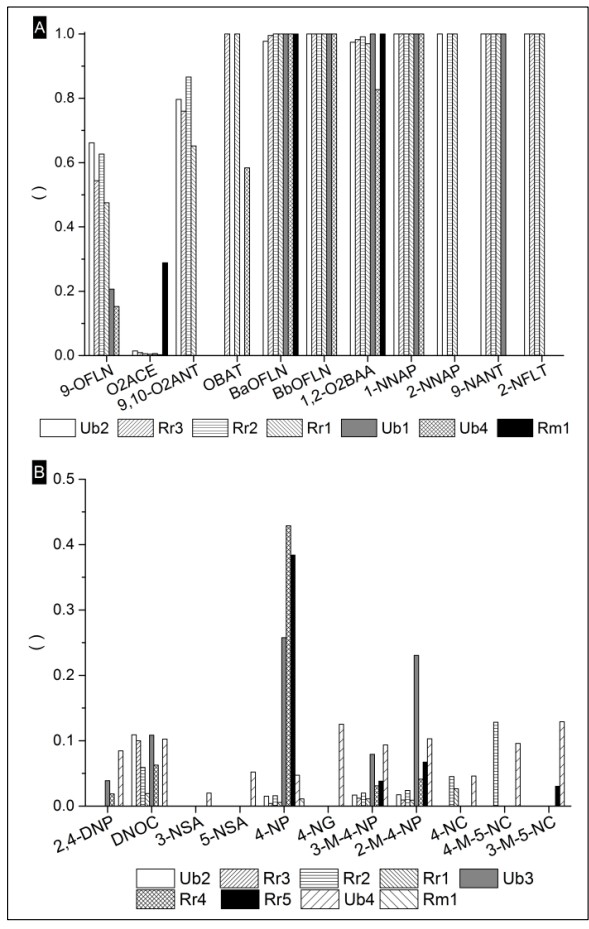