# Peer review of "Snow scavenging and phase partitioning of nitrated and oxygenated aromatic hydrocarbons in polluted and remote environments in central Europe and the European Arctic"

_Atmospheric Chemistry and Physics, 2017_

## Referee Comment (RC1) · Anonymous Referee #3 · 23 Feb 2018

**GENERAL COMMENTS**

The authors present a valuable dataset on the content of harmful nitro(poly)aromatic pollutants in snow precipitation. The results of field samples analyses are compared with the theoretically predicted particulate mass fractions based on substance gas-particle partitioning constants and driving forces of semi-volatile organic compounds scavenging by the atmospheric condensed phases (particulate and liquid) are discussed. Sampling, sample preparation and handling, and the subsequent analytical procedures were well-planned. I find a few shortcomings in the data analysis and/or presentation (see specific comments). Phase partitioning is addressed at the end of the manuscript, however for the sake of clarity I would like to see its indication already in the Introduction section. When considering PM with a water layer, there are two interphases with the corresponding partitioning/equilibria that have to be taken into account – gas-liquid and solid-liquid. To avoid confusion, the experimental system needs to be defined at the beginning of the manuscript. Overall, presentation of the data is superficial, confusing and even unclear in some parts, and this aspect of the manuscript should be substantially improved before publication (see specific comments). I also encourage the authors to edit the language of the manuscript thoroughly. I think this is a very important dataset that should be delivered to the atmospheric science community, however the presentation and discussion should be improved; I suggest a major revision of the manuscript before publication.

**SPECIFIC COMMENTS**

I was confused with the introduction of particle scavenging and gas scavenging in the Introduction section. Can you correlate $c_p$ and $c_{pp}$? From my point of understanding it should be the same value for low-solubility compounds (NPAH and OPAH). For water soluble compounds (NMAH), $c_{pp}+c_{pd}$ value should be equal (for high $\theta$) or bigger (for low $\theta$) than $c_p$. Additional explanation would help understand the investigated multiphase system.

Please rephrase the sentence on P3L16-18 starting with 'However, it is not known if this concept also applies to hydrophilic SOCs which may also demonstrate high $\theta$,…' – I would say it can be speculated or is expected that water solubility plays a role in wet scavenging rather than 'not known' as stated above.

I want to comment on the LOQ and the precision of the reported data. First of all, it seems that your LOQ was lower than the lowest concentration used for the calibration curve. This is a bad practice. Whenever one estimates LOQ from IDL, this should be verified with the calibration curve. Then, I could only calculate ambient concentrations from the sample concentrations for N/OPAH as no final sample volume is reported for NMAH. Please add it and verify the LOQ you set. Second, you don't report on the precision of the analytical procedure/measurement. Please add the level of confidence to the data and report the concentrations accordingly (so far almost all concentrations ranging from 0.09 to thousands are reported with 2 decimal places, which is hard to believe it).

The advantage of air mass trajectories is not clear to me. You don't use them in the discussion of source apportionment, which is a bit superficial in general. I am aware that this was not the main scope of the study, but if you discuss possible sources of particular pollutants, you should combine

the knowledge with the estimated air mass trajectories and this should bring you some conclusions. If you don't want to go deeper here, I would shorten the source apportionment parts in the Results and discussion section (P8L15-23 and P8L31-P9L3). I also don't know what this means: 'The snowfalls leading to samples Ub1, Ub3, Rr1, and Rr5 followed immediately frontal passages with advection from westerly directions (Fig. S2), unlike in the other precipitation events.' Could you add an explanation?

I would move the last paragraph of the Results and discussion to the beginning of the section (to start with less sophisticated samples). I am also not sure if there is a need to report all values here, as they are also shown in Fig. 1. The same applies to the next section (NMAH) and Fig. 2.

The acronym SOC in not intuitive for me, I would rather suggest the use of SVOC for Semi Volatile Organic Compounds.
* * *
P1L9: You use $ng\ L^{-1}$ for ΣNPAH and $\mu g$ L-1 for ΣOPAH, which is misleading. Unify the units all over the manuscript.

P1L14: 'The lowest levels of ΣOPAHs and ΣNMAHs were found at the remote site (9.2 and 390.5 ng L-1, respectively).' – what about NPAHs? You should comment on their concentrations as well in the abstract.

P1L18-19: 'interplay between gas-particle partitioning in the aerosol, particle mass size distribution, and dissolution during in- or below-cloud scavenging.' – particle mass size distribution was not clear for me, maybe 'PM size-dependent mass distribution'? The same applies to P13L3.

P2L12: 'NPAHs are also formed through reactions in the aerosol condensed phase (Keyte et al., 2013; Jariyasopit et al., 2014).' – why didn't you measure them in the dissolved phase? Please comment.

P5L10: As no other glassware was pre-baked, I wonder if there is a reason why pre-baked glass inserts were used.

P5L23: 'internal method' – do you mean internal standard method?

P6L23: '(NH4)2SO4 and NaCl (the last two represent secondary inorganic aerosols)' – why NaCl secondary?

P8L4-14: I suppose this paragraph reports only the values for dissolved phase. This is not clear from the sentence starting in L7 on. Please clarify.

P9L11-12: Reference is missing.

P10L8-11: in some samples 4-NG was not detected; start the sentence with 'When detected,…'

P10L35-P11L1: this is not true – nitro group is strongly e-withdrawing (acceptor), therefore lower e-density on the aromatic ring (lower e-donor ability) and weaker π-interactions. Additional $NO_2$ group on the aromatic ring increases the number of possible hydrogen bonds. However, as you conclude correctly, intramolecular H-bond between adjacent -OH and $-NO_2$ opposes, which results in lesser interactions with the surrounding molecules for DNP in comparison to NP.

P11L5-12: Comment also low predicted θ values of 1,4-O$_2$NAP, 1-NNAP, 2-NNAP, 5-NACE, 2-NFLN.

P11L15: define W$_T$

P11L23-25: Can you definitely exclude post-sampling dissolution of water-soluble compounds in the liquid phase?
* * *
DATA PRESENTATION

There is a long list of measured compounds, but not so many are shown in the graphs. It took me quite some time to compare Table 2 (compound name & abbreviation), Fig. 1 (measured concentration in each fraction – denoted with abbreviations) and Table S4A (measured sum concentrations – denoted with long names). I would suggest adding a column or two in Table 2 and mark whether the compound was detected in particulate, dissolved, or both phases; or if it was not detected at all. Besides, 2-Nitro-9-fluorenone is listed among OPAHs in Table 2 – is this on purpose?

Table S1: In the fifth column you report 'Total NMAHs', but then you use superscripts TOT, P, D within the table. This should be clarified and explain the superscripts.

Abbreviations should also be introduced in the SI – maybe add them to Table S2.

Table S4A and B: add confidence intervals and report the data accordingly.

Table S4B: how did you convert $c_{pd+pp}$ (ng l$^{-1}$) into $c_{pd+pp}$ (ng m$^{-3}$)? Explain θ in the footnote.

**TECHNICAL CORRECTIONS**

Only a few technical shortcomings are mentioned here. I suggest a thorough English-proofing before publication.

P1L3: 'Their precipitation cycling has hardly been studied.' – wet deposition by precipitation is a part of environmental cycling, I don't understand the meaning of *precipitation cycling*, please correct.

P1L10: 'snow dissolved and particulate phase' – 's' is missing in phases.

P3L22: 'dinitrophenols (2,4-dinitrophenol (2,4-DNP) and 2-methyl-4,6-dinitrophenol (i.e. dinitro-ortho-cresol, DNOC) were the most frequently measured nitrophenols…' – end-bracket is missing after DNOC) ant present perfect should be used afterwards (i.e. DNOC)) have been…).

P5L15: 30m – space is missing

P5L17: 15 °C – erase the space

P7L1: 'fOM (the mixing ratio of total organic matter in PM) × 0.60 and fOM×0.40' – all spaces or no spaces between the symbol, ×, and the number

P7L5: use past tense in the sentence starting with 'The individual partitioning…'

---

## Referee Comment (RC2) · Anonymous Referee #2 · 4 Apr 2018

Shahpoury et al. present data on nitrated and oxygenated polycyclic aromatic hydrocarbons (N/OPAHs) from snow samples at different European sites. They report concentrations from particulate and liquid (melted) snow phases and estimate the fraction of N/OPAHs removed by precipitation through particle scavenging based on predicted particulate mass fractions and observed snow phase partitioning. Such data are sparse and could in principle be helpful to better understand wet removal of hydrophobic and less hydrophobic organic compounds from the atmosphere. I have, however, a number of major issues with the applied methodology and data processing which need

clarification before the paper can be considered for publication. In addition, the paper lacks important details and explanations to be able to fully understand what has been done. The structure of the paper and the clarity of the results' presentations need also be improved. I recommend re-consideration after major revision.

1) The authors aim at measuring the phase partitioning of N/OPAHs in snow by analyzing both the snow particulate and aqueous phases. To do so, they pass the melted snow samples through a filtration-extraction system. My concern is that the observed distributions between snow particulates (retained on a filter) and snow water (extracted on a sorbent) might be strongly biased for compounds with some water solubility, which upon sample thawing will dissolve from the particulates into the melt water. This potential artefact needs to be thoroughly addressed as it might render many of the presented results useless. One of the main findings of the study is the unexpected behavior of highly water soluble NMAHs (p11 l29-30). Considering the potential experimental bias, I wonder if this finding really holds.

2) Snow particulate concentrations are given in ng L-1, which will strongly depend of the final volume during sample extraction. As this final volume is an arbitrary choice of the authors (and not even reported), it is not clear to me how these concentrations can be used in any reasonable way beyond comparing between different samples. Even the comparison with snow aqueous phase concentrations seems difficult to me. Adding them up to a total snow concentration, as done in Fig. 1 and Table S4A seems hard to justify as well to me.

3) To make my confusion complete, in Table S4B the authors calculate dimensionless total scavenging ratios from the sum of snow particulate and aqueous phase (in ng L-1) and the sum of particle and gas phase concentrations (in ng m-3), obtained just before the snow events. This ratio seems to be derived by multiplying ng L-1 concentrations by a factor of 1000 and then dividing by ng m-3 concentrations, ignoring that the denominator in the unit is referring to solvent volume in one case and air volume in the other. This seems very odd to me and needs explanation.

4) The conclusion that the phase partitioning of NMAHs is determined by an interplay between GPP, particle mass size distribution, and dissolution during in- or below-cloud scavenging (abstract l17-19 and p12 l13-19) is weak and not convincing. Even assuming the applied methodology was artefact-free and the calculation of the total scavenging ratio can be justified (see above), I cannot follow the authors' reasoning why the observations would indicate an importance of particle mass size distribution and dissolution. This needs to be much better laid out in the discussion and all data in support need to be shown (p12 l14-15). Much of this conclusion seems to originate from one single sample, where additional measurements have been made. The poor robustness of results that would follow from this needs to be addressed as well.

5) On p7 l13ff and p12 l6ff the authors discuss acid dissociation in particles in relation to pH. However, they seem to not be aware of sample pH being a different thing than particle pH, i.e. pH of particle liquid water. With a melted snow sample being highly diluted in comparison to aerosol particles, the measured sample pH cannot be used to describe acid dissociation in particle liquid water. This needs to be corrected appropriately.

6) A "dissolved" phase is not a thing. There are particulate, aqueous, or gas phases, for example, but a dissolved phase does not exist and the term needs to be replaced throughout all the manuscript.

7) The experimental section lacks many details, e.g. flow rate during extraction (p4 l23), volume of ethyl acetate (p5 l8-9), volume of buffer (p5, l32-33), concentration of EDTA (p5, l33), duration of ultrasonication (p6 l1), volume of buffer (p6 l1-2), etc. Please make sure any reader would be able to fully repeat your experiments with the information given.

8) Data presentation could be improved. All the abbreviations for the different sites are impossible to remember. I suggest removing them completely from the discussion text and referring to the name of the sites instead. Other abbreviations are used without

explanation, e.g. the indices WSOM and OP (p6, l29). Different panels in the Figures are sometimes difficult to compare, because the order of compounds changes. Figure 3 contains empty brackets in the y-axis label, Figure 4 even contains only empty brackets as y-axis label. The discussion of air mass back trajectories in 3.1 is odd, as it is nowhere else in the manuscript referred to. If still important, it should be substantiated.

9) Correction factors for fOM (p7 l1-2) are taken from a study in the 1980s done in Los Angeles, USA. I wonder if this is really the best reference for the sites sampled by the authors. Also, these factors are likely to differ substantially both between sites but also between different meteorological situations. A note on the sensitivity of the results on such highly uncertain parameters would be helpful.

10) Sections 3.2 and 3.3 are tedious to read with all the abbreviations and very detailed concentrations. I suggest discussing the main observations in these measurements in a more compact way and leaving much of the numbers to the corresponding Figures and Tables. Also, some paragraphs might be moved to the Introduction (e.g. p8 l14-22). The space gained here should be used to discuss main findings of the paper in a more substantiated way (see comment above).

---

## Author Response (AR1)

**Authors' Responses to Reviewers' Comments**

Journal: Atmospheric Chemistry and Physics

Manuscript ID: acp-2017-1242

**Manuscript Title:** Snow scavenging and phase partitioning of nitrated and oxygenated aromatic hydrocarbons in polluted and remote environments in central Europe and the European Arctic **Authors:** Pourya Shahpoury, Zoran Kitanovski, Gerhard Lammel

**Reviewer #1 Comments**

**1) Reviewer's comment**

The authors present a valuable dataset on the content of harmful nitro(poly)aromatic pollutants in snow precipitation. The results of field samples analyses are compared with the theoretically predicted particulate mass fractions based on substance gas-particle partitioning constants and driving forces of semi-volatile organic compounds scavenging by the atmospheric condensed phases (particulate and liquid) are discussed. Sampling, sample preparation and handling, and the subsequent analytical procedures were well-planned. I find a few shortcomings in the data analysis and/or presentation (see specific comments).

**Authors' response**

We appreciate the reviewer's careful consideration of our manuscript and constructive comments. We have addressed the reviewer's specific comments below.

**2) Reviewer's comment**

Phase partitioning is addressed at the end of the manuscript, however for the sake of clarity I would like to see its indication already in the Introduction section. When considering PM with a water layer, there are two interphases with the corresponding partitioning/equilibria that have to be taken into account – gas-liquid and solid-liquid. To avoid confusion, the experimental system needs to be defined at the beginning of the manuscript.

**Authors' response**

We have added a statement regarding the partitioning system that we considered for estimating the particulate mass fractions of our target substances:

Page 4, L23-26: "For calculating  $\Theta$ , our method took into account the interaction of atmospheric SVOCs with PM liquid organic and polymeric phases, as well as the interaction with PM black carbon and salts, while disregarding the partitioning into PM aqueous phase, particle-liquid interactions, and liquid-liquid phase separation within PM (Sect. 2.6)."

**3) Reviewer's comment**

Overall, presentation of the data is superficial, confusing and even unclear in some parts, and this aspect of the manuscript should be substantially improved before publication (see specific comments). I also encourage the authors to edit the language of the manuscript thoroughly. I think this is a very important dataset that should be delivered to the atmospheric science community, however the presentation and discussion should be improved; I suggest a major revision of the manuscript before publication.

**Authors' response**

We have improved data presentation by re-making the graphs, now with better graphics and more intuitiveness. For added clarity, we replaced the sample site codes with those that reflect the site names. In addition, we have focused on describing the study's major findings and removed description of less significant results from the text. Please see below our responses to specific comments.

**4) Reviewer's comment**

I was confused with the introduction of particle scavenging and gas scavenging in the Introduction section. Can you correlate cp and cpp? From my point of understanding it should be the same value for low-solubility compounds (NPAH and OPAH). For water soluble compounds (NMAH), cpp+cpd value should be equal (for high  $\theta$ ) or bigger (for low  $\theta$ ) than cp. Additional explanation would help understand the investigated multiphase system.

**Authors' response**

For further clarification, we have modified the statement as follows:

Page 3, L34 - Page 4, L6: "...i.e. the higher the  $\Theta$  at a given temperature, the more efficient is the scavenging, and the magnitude of  $\Theta_W$  is expected to be close to  $\Theta$ . However, it is anticipated that for hydrophilic SVOCs, which may demonstrate low or high  $\Theta$ , water solubility plays an additional role in the substance wet scavenging pathways and the comparability of  $\Theta_W$  with  $\Theta$ . For instance, a NMAH with high  $\Theta$  in the atmosphere may demonstrate low  $\Theta_W$  due to substance dissolution in hydrometeors following particle scavenging. Conversely, a NMAH with low  $\Theta$ is expected to show low  $\Theta_W$ , as it undergoes gas scavenging process."

**5) Reviewer's comment**

Please rephrase the sentence on P3L16-18 starting with 'However, it is not known if this concept also applies to hydrophilic SOCs which may also demonstrate high  $\Theta, ..., -I$  would say it can be speculated or is expected that water solubility plays a role in wet scavenging rather than 'not known' as stated above.

**Authors' response**

We have changed the sentence as follows:

Page 4, L2: "However, it is anticipated that for hydrophilic SVOCs, which may demonstrate low or high  $\Theta$ , water solubility plays...."

**6) Reviewer's comment**

I want to comment on the LOQ and the precision of the reported data. First of all, it seems that your LOQ was lower than the lowest concentration used for the calibration curve. This is a bad practice. Whenever one estimates LOQ from IDL, this should be verified with the calibration curve. Then, I could only calculate ambient concentrations from the sample concentrations for N/OPAH as no final sample volume is reported for NMAH. Please add it and verify the LOQ you set. Second, you don't report on the precision of the analytical procedure/measurement. Please add the level of confidence to the data and report the concentrations accordingly (so far almost all concentrations ranging from 0.09 to thousands are reported with 2 decimal places, which is hard to believe it).

**Authors' response**

- This is not correct – measured concentrations which were smaller than the lowest calibration point (IDL) were always discarded and never used. The limit of quantification (LOQ) with our method cannot possibly be lower that IDL because our LOQs are defined as the mean concentration of analytes in field blanks + 3 standard deviations, and where analyte concentrations in blanks were <IDL, we replaced them with IDL. The reviewer's misunderstanding may be related to the sentence "...IDL values were used in cases where analyte concentrations in blanks were <IDL." For clarification, we have revised the sentence as follows:

Page 7, L13-15: "LOQ values were determined as mean concentration of each analyte in blanks + 3 standard deviations. For this purpose, blank values that were <IDL were replaced with IDL values. Where analyte concentrations in samples exceeded the LOQ, mean blank concentrations were subtracted from those in the corresponding samples."

- It is not clear to us what the reviewer implies by "*calculate ambient concentrations from the sample concentrations*". We intended to report analyte concentrations in snowmelt and that is what we did. Back-calculating ambient concentrations was not the intention of our study. For clarification, we have now added snowmelt volumes for each sample: Page 20, Table 1 (last column)
- We don't think it is correct to report confidence levels for our dataset because we do not report means for analysis of replicate samples; in our study n = 1 per site. In order to address the reviewer's concern, we have now reported the concentrations >1 ng L-1 with one decimal place and those smaller than 1 ng L-1 with two decimal places (or more in exceptional cases in Table S4B) throughout the text and the Supplement.

**7) *Reviewer's comment**

The advantage of air mass trajectories is not clear to me. You don't use them in the discussion of source apportionment, which is a bit superficial in general. I am aware that this was not the main scope of the study, but if you discuss possible sources of particular pollutants, you should combine the knowledge with the estimated air mass trajectories and this should bring you some conclusions. If you don't want to go deeper here, I would shorten the source apportionment parts in the Results and discussion section (P8L15-23 and P8L31-P9L3). I also don't know what this means: 'The snowfalls leading to samples Ub1, Ub3, Rr1, and Rr5 followed immediately frontal passages with advection from westerly directions (Fig. S2), unlike in the other precipitation events.' Could you add an explanation?

**Authors' response**

- We have revised and moved the discussion of the air mass history analysis (previously Section 3.1) to the Supplement:

Page S5 in the Supplement, Text S1: "For all central European sites, the air masses corresponding to the snow samples had been advected mostly from westerly directions (Figure S2). They had passed over polluted areas of central and western Europe (such as in NE France/SW Germany, W and SE Germany for samples MZ15, OS, WB and AB) until the last 100-200 km before precipitation started, when they had been transported over rural areas. In conclusion, these air mass histories are quite typical for the region in the sense that the source areas of the region have contributed to pollution loading, while point sources or a single source area, such as a close city, was never determining the loading. The snowfalls leading to samples MZ15, OS, WB, and PP2 immediately followed frontal passages with advection from westerly directions (trajectories are shown in Figure S2), unlike in the other precipitation events. Snow fall in air masses following a frontal passage may have been on-going for some time prior to arrival. This could lead to somewhat lower concentrations in precipitation, as the gases and particles, to be eventually transferred into snow, may had been previously scavenged."

- We have added a new sentence under the current Section 3.1, i.e. "*N/OPAH concentrations and distribution in snow*", (formerly Section 3.2) with reference to the air mass histories description in the Supplement.

Page 9, L8-11: "9-OFLN, O2ACE, and 9,10-O2ANT originate from both primary (e.g. diesel exhaust, coal and biomass burning) and secondary sources, whereas OBAT and BaOFLN are associated with primary sources (see references in Sect. 1). Potential source areas for such emissions are reflected in the air mass histories of all samples (see Text S1 and Fig. S2)."

**8) Reviewer's comment**

I would move the last paragraph of the Results and discussion to the beginning of the section (to start with less sophisticated samples). I am also not sure if there is a need to report all values here, as they are also shown in Fig. 1. The same applies to the next section (NMAH) and Fig. 2.

**Authors' response**

It is not clear to us what part of the results and discussion the reviewer is referring to (i.e. under which section). However, we have moved the discussion of pollution sources for OPAHs from Section 3.1 (previously 3.2) to Section 1 (Introduction) following a suggestion made by the second reviewer; however, in Section 3.1, we have made a short note about potential emission sources of OPAHs:

Page 2, L17-29: "For instance, 9-fluorenone (9-OFLN), 9,10-anthraquinone (9,10-O2ANT), 1,4-naphthoquinone (1,4-O2NAP), and 1,2-benzanthraquinone (1,2-O2BAA) were previously found in diesel exhaust (Choudhury, 1982; Cho et al., 2004) and biomass ...... For instance, formation of 1,4-O2NAP and 9,10-O2ANT following photolysis of 1-nitronaphthalene (1-NNAP) and 9-nitroanthracene (9-NANT) was suggested by previous studies (Keyte et al., 2013; Bandowe and Meusel, 2017)."

Page 9, L8-10: "9-OFLN, O2ACE, and 9,10-O2ANT originate from both primary (e.g. diesel exhaust, coal and biomass burning) and secondary sources, whereas OBAT and BaOFLN are associated with primary sources (see references in Sect. 1)."

In addition, following the suggestions by both reviewers, we have now highlighted more important findings and we removed less significant results from Section 3.1 and 3.2 in the current edition (formerly, Section 3.2 and 3.3):

**9) *Reviewer's comment**

The acronym SOC in not intuitive for me, I would rather suggest the use of SVOC for Semi Volatile Organic Compounds.

**Authors' response**

SOC has been replaced with SVOC throughout the text.

Page 3, L18, L20, L21, L23, L24, L28, L29, L33, L34 Page 4, L2, L7, L23

**10) Reviewer's comment**

P1L9: You use ng L-1 for  $\Sigma$ NPAH and  $\mu$ g L-1 for  $\Sigma$ OPAH, which is misleading. Unify the units all over the manuscript

**Authors' response**

The units have been unified:

Page 1, L8-10: " $\sum NPAH$  concentrations were 1.2-17.6 and 8.8-19.1 ng  $L^{-1}$  at urban and rural sites, whereas  $\sum OPAHs$  were 269.5 – 1114.1 and 478.7 - 2384.4 ng  $L^{-1}$  at these sites, respectively."

**11) *Reviewer's comment**

P1L14: 'The lowest levels of  $\Sigma$ OPAHs and  $\Sigma$ NMAHs were found at the remote site (9.2 and 390.5 ng L-1, respectively).' – what about NPAHs? You should comment on their concentrations as well in the abstract.

**Authors' response**

It has been revised: Page 1, L14-15: "The lowest levels of  $\sum N/OPAHs$  and  $\sum NMAHs$  were found at the remote site..."

**12) Reviewer's comment**

P1L18-19: 'interplay between gas-particle partitioning in the aerosol, particle mass size distribution, and dissolution during in- or below-cloud scavenging.' – particle mass size distribution was not clear for me, maybe 'PM size-dependent mass distribution'? The same applies to P13L3.

**Authors' response**

It has been revised:

Page 1, L18-19: "...i.e. NMAHs, is determined by an interplay between gas-particle partitioning in the aerosol and dissolution during in- or below-cloud scavenging."

**13) *Reviewer's comment**

P2L12: 'NPAHs are also formed through reactions in the aerosol condensed phase (Keyte et al., 2013; Jariyasopit et al., 2014).' – why didn't you measure them in the dissolved phase? Please comment.

**Authors' response**

The following statement has been added under Section 3.1 in order to address the reviewer's comment:

Page 9, L12-17: "NPAHs were not found in the snow aqueous phase. Our GPP model suggests that at near-zero temperatures the targeted NPAHs would be completely sorbed to the particulate phase in the atmosphere, with the exception 1-NNAP, 2-NNAP, 5-NACE, and 2-NFLU, which would partition between gas and particulate phases. Regardless, relatively low water solubility of NPAHs, indicated by their octanol-water partitioning coefficients (log  $K_{OW}$ ; Fig. S3), may limit their gas scavenging from the atmosphere and subsequent presence in the snow aqueous phase."

**14) *Reviewer's comment**

P5L10: As no other glassware was pre-baked, I wonder if there is a reason why pre-baked glass inserts were used.

**Authors' response**

We have revised the sentence as follows:

Page 6, L7-8: "...transferred to 2-mL vials containing pre-baked 0.4-mL glass inserts for further analysis. All other glassware used for sample analysis were washed with lab-grade detergent and deionized water, and baked at 300°C for 12hrs."

**15) *Reviewer's comment**

P5L23: 'internal method' - do you mean internal standard method?

**Authors' response**

It has been revised as follows:

Page 6, L21-22: "The analyte quantification was done using the internal calibration method with 11-point calibration curves..."

**16) **Reviewer's comment**

P6L23: '(NH4)2SO4 and NaCl (the last two represent secondary inorganic aerosols)' – why NaCl secondary?

**Authors' response**

The statement has been revised as follows:

Page 7, L22-23: "...( $NH_4$ )2SO4 and NaCl (the last two represent the contributions of secondary inorganic aerosols and sea salt) ..."

**17) Reviewer's comment**

P8L4-14: I suppose this paragraph reports only the values for dissolved phase. This is not clear from the sentence starting in L7 on. Please clarify.

**Authors' response**

We have added the phrase "aqueous" for clarification:

Page 9, L3-4: "In the aqueous phase, 9-OFLN, O2ACE, and 9,10-O2ANT were found in nearly all samples..."

Page 9, L5: "O2ACE was the most abundant in the aqueous phase..."

Page 9, L7: "...BaOFLN were found less frequently with relatively low concentrations in the aqueous phase..."

Page 9, L11: "Overall, TF and MZ17 were the least and most polluted sites, with  $\sum OPAH$  aqueous concentrations..."

**18) *Reviewer's comment**

P9L11-12: Reference is missing

**Authors' response**

The related references have been added: Page 9, L36-37: "...with previous findings in the literature (Albinet et al., 2007; Souza et al., 2014; Lin et al., 2015; Tomaz et al., 2016)."

**19) *Reviewer's comment**

P10L8-11: in some samples 4-NG was not detected; start the sentence with 'When detected...'

**Authors' response**

Revised as suggested.

Page 10, L21-22: "When detected, the aqueous phase concentrations of 4-NG were comparable in urban and rural samples"

**20) Reviewer's comment**

P10L35-P11L1: this is not true – nitro group is strongly e-withdrawing (acceptor), therefore lower density on the aromatic ring (lower e-donor ability) and weaker  $\pi$ -interactions. Additional NO2 group on the aromatic ring increases the number of possible hydrogen bonds. However, as you conclude correctly, intramolecular H-bond between adjacent -OH and -NO2 opposes, which results in lesser interactions with the surrounding molecules for DNP in comparison to NP.

**Authors' response**

The reviewer is referring to pi-pi interaction which exist between neighboring electron-deficient and electron-rich aromatic rings, whereas we commented on H-bonding in the manuscript, i.e. interactions between neighboring H-donor and e-donor molecules. We have now revised the sentence for clarification:

Page 11, L6-8: "the presence of two nitro groups on 2,4-DNP is expected to promote stronger H-bonding with PM, compared to 4-NP which has one nitro group (compare the Abraham descriptor B for the two compounds in Table S2)."

Page 11, L11-12: "This reduces the H-bonding ability of 2,4-DNP compared to 4-NP (compare the Abraham descriptor A in Table S2)."

**21) Reviewer's comment**

P11L5-12: Comment also low predicted θ values of 1,4-O2NAP, 1-NNAP, 2-NNAP, 5-NACE, 2-NFLN.

**Authors' response**

We have revised the entire paragraph and included explanation for phase partitioning of 1,4-O2NAP, 1-NNAP, 2-NNAP. The other two substances that the reviewer is referring to (5-NACE and 2-NFLU) were not found in our study or were below LOQs, hence, no comments has been made about them.

Page 11, L13-25: "For substances which demonstrate complete sorption to PM, particle scavenging is expected to be the dominant source of wet deposition and, consequently, such substances will be enriched in precipitation particulate phase. Our observations depicted in Fig. 7A support this assumption, namely, N/OPAHs with high  $\Theta$ (Fig. 6A-B) were largely associated with precipitation particulate phase. The only exception was 9,10-O2ANT in MZ15 and MZ17 samples. For substances that distribute between gas and particulate phases, both gas and particle scavenging are relevant; however, the substance water solubility is a factor that could enhance or limit the gas scavenging mechanism, regardless of the compound phase partitioning in the atmosphere. For instance, 1- and 2-NNAP are expected to be  $\geq$ 90% in the gas phase under our experimental conditions; nonetheless, they were found in the precipitation particulate phase. A similar effect was seen for 9-OFLN. This is explained by the substances' relatively low water solubility (see estimated KOW in Fig. S3) which limits their dissolution in hydrometeors. On the contrary, 1,4-O2NAP ( $\Theta$ : 0.001- 0.003) and O2ACE ( $\Theta$ : 0.24- 0.56) were more enriched in the aqueous phase, which is explained by their relatively low log KOW(1.95-2.13; Fig. S3). Overall, our findings support the implied assumption that phase partitioning in air is preserved in snow, provided that the substance solubility in water is not a limiting factor."

22) *Reviewer's comment* P11L15: define WT

**Authors' response**

We have added a new equation under Introduction, Eq. 3, and introduced the definition of  $W_{\rm T}$ :

Page 4, L7-10: "The efficiency of SVOC wet scavenging is defined by  $W_T$  (unitless) (Škrdlíková et al., 2011; Shahpoury et al., 2015) Eq. (3):

 $W_{\rm T} = (c_{\rm snow} \times 1000)/c_{\rm air}$

where  $c_{snow}$  is the total analyte concentration in snow (ng  $L^{-1}$ ) and  $c_{air}$  is that (ng  $m^{-3}$ ) in ambient air."

**23) Reviewer's comment**

P11L23-25: Can you definitely exclude post-sampling dissolution of water-soluble compounds in the liquid phase?

**Authors' response**

We rule out post-sampling dissolution of target compounds due to reasons mentioned below - we have added the following statements to the text in order to address the reviewer's concern:

Page 5, L11-15: "Our sample processing was performed in such way to minimize the analyte phase change prior to analysis, namely the samples were thawed at room temperature in the lab and, immediately after thawing, when the samples were near freezing point, the meltwater was passed through a pre-assembled filtration-extraction setup (Fig. S1), which allowed simultaneous separation of meltwater particulate phase and extraction of aqueous phase. This made it possible to minimize the time that particles were in contact with the aqueous phase of meltwater."

Page 11, L31-35: "We rule out post-sampling dissolution of NMAHs from particulate to aqueous phase for the reasons mentioned in Sect. 2.2. In addition, looking at the Czech samples, 3-M-4-NP and 2-M-4-NP, which have lower water solubility (log  $K_{OW}$ : 2.27-2.43; Fig. S3) than 4-NP (log  $K_{OW}$ : 1.68) and comparable  $pK_a$  values (7.15-7.33), demonstrated much higher partitioning in the aqueous phase than 4-NP did. If a post-sampling phase change had occurred, we would have observed an opposite pattern."

**24) Reviewer's comment**

There is a long list of measured compounds, but not so many are shown in the graphs. It took me quite some time to compare Table 2 (compound name & abbreviation), Fig. 1 (measured concentration in each fraction – denoted with abbreviations) and Table S4A (measured sum concentrations – denoted with long names). I would suggest adding a column or two in Table 2 and mark whether the compound was detected in particulate, dissolved, or both phases; or if it was not detected at all. Besides, 2-Nitro-9-fluorenone is listed among OPAHs in Table 2 – is this on purpose?

**Authors' response**

We have made the suggested changes in Table 2:

Page 21, Table 2, last column. In addition, we think that 2-nitro-9-fluorenone should be placed under OPAHs; no changes made.

**25) Reviewer's comment**

Table S1: In the fifth column you report 'Total NMAHs', but then you use superscripts TOT, P, D within the table. This should be clarified and explain the superscripts.

**Authors' response**

We have now added the missing legends to Table S1:

Page S2, Table S1 caption: "a: including nitrophenols, methylnitrophenols and dinitrophenols; b: including 4-NC, 3-M-5-NC and 4-M-5NC; pp: precipitation particulate phase; pa: precipitation aqueous phase; TOT: total

concentration in precipitation (particulate + aqueous phase concentration); N.A.: data not available; \*: below limit of detection (LOD); \*\*: below limit of quantification (LOQ); \*\*\*: median concentration"

**26) Reviewer's comment**

Abbreviations should also be introduced in the SI – maybe add them to Table S2.

**Authors' response**

The abbreviations have been added:

Page S3 (Table S2, second column) and S8-S10 (Table S4A and S4B, second column) in the Supplement.

**27) Reviewer's comment**

Table S4A and B: add confidence intervals and report the data accordingly.

**Authors' response**

As previously noted in response to reviewer's comment 6, we do not think it is correct to report confidence intervals (please see our response above). However, we are now reporting values >1 ng L-1 with one decimal place and those <1 ng L-1 with two decimal places, or more in exceptional cases in Table S4B.

**28) Reviewer's comment**

Table S4B: how did you convert cpd+pp (ng l-1) into cpd+pp (ng m-3)? Explain  $\theta$  in the footnote.

**Authors' response**

The formula in Table S4B caption has been modified to reflect the conversion from ng L-1 to ng m-3. The formula for calculating  $\theta$  has also been added to Table S4B caption:

Page S10, Table S4B: "...WT: total scavenging ratio (dimensionless) =  $[c_{pa+pp} (ng L^{-1}) \times 1000]/ [c_{g+}c_p (ng m^{-3})]$ ..." "... $\Theta$ : particulate mass fraction =  $c_p/(c_g + c_p)$ ..."

**29) Reviewer's comment**

P1L3: 'Their precipitation cycling has hardly been studied.' – wet deposition by precipitation is a part of environmental cycling, I don't understand the meaning of precipitation cycling, please correct.

**Authors' response**

We have revised the sentence as follows: Page 1, L3: "Their environmental cycling through wet deposition has hardly been studied."

**30) Reviewer's comment**

P1L10: 'snow dissolved and particulate phase' - 's' is missing in phases

**Authors' response**

It has been revised:

Page 1, L10-11 "Acenaphthoquinone and 9,10-anthraquinone were predominant in snow aqueous and particulate phases, respectively."

**31) Reviewer's comment**

P3L22: 'dinitrophenols (2,4-dinitrophenol (2,4-DNP) and 2-methyl-4,6-dinitrophenol (i.e. dinitroortho-cresol, DNOC) were the most frequently measured nitrophenols...' – end-bracket is missing after DNOC) ant present perfect should be used afterwards (i.e. DNOC)) have been...).

**Authors' response**

The statement has been revised as follows:

Page 4, L12-15: "...4-nitrophenol (4-NP), several methyl-nitrophenol isomers as well as dinitrophenols, 2,4dinitrophenol (2,4-DNP) and 2-methyl-4,6-dinitrophenol (i.e. dinitro-ortho-cresol, DNOC), have been the most frequently measured nitrophenols in precipitation in urban and rural Europe..."

32) *Reviewer's comment* P5L15: 30m – space is missing

**Authors' response**

Revised:

Page 6, L13: "...column (30 m + 10 m integrated guard..."

33) *Reviewer's comment* P5L17: 15 °C – erase the space

**Authors' response**

Revised:

Page 6, L15: "... then ramped to 180°C at 15°C..."

**34) Reviewer's comment**

P7L1: 'fOM (the mixing ratio of total organic matter in PM)  $\times$  0.60 and fOM $\times$ 0.40' – all spaces or no spaces between the symbol,  $\times$ , and the number

**Authors' response**

Revised:

Page 7, L29: "...corresponding to  $f_{OM}$  (the mixing ratio of total organic matter in PM) × 0.60 and  $f_{OM}$  × 0.40..."

35) *Reviewer's comment*P7L5: use past tense in the sentence starting with 'The individual partitioning...

**Authors' response**

Revised:

Page 8, L7: "... The individual partitioning constants used in the multi-phase model were calculated..."

**Reviewer #2 Comments**

**Reviewer's general comment**

Shahpoury et al. present data on nitrated and oxygenated polycyclic aromatic hydrocarbons (N/OPAHs) from snow samples at different European sites. They report concentrations from particulate and liquid (melted) snow phases and estimate the fraction of N/OPAHs removed by precipitation through particle scavenging based on predicted particulate mass fractions and observed snow phase partitioning. Such data are sparse and could in principle be helpful to better understand wet removal of hydrophobic and less hydrophobic organic compounds from the atmosphere. I have, however, a number of major issues with the applied methodology and data processing which need clarification before the paper can be considered for publication. In addition, the paper lacks important details and explanations to be able to fully understand what has been done. The structure of the paper and the clarity of the results' presentations need also be improved. I recommend re-consideration after major revision.

**Authors' response**

We thank the reviewer for their thorough review of our manuscript and for valuable comments and suggestions. Below, we have addressed the reviewer's specific comments regarding our analytical methods as well as data processing and presentation.

**1) Reviewer's comment**

The authors aim at measuring the phase partitioning of N/OPAHs in snow by analyzing both the snow particulate and aqueous phases. To do so, they pass the melted snow samples through a filtration-extraction system. My concern is that the observed distributions between snow particulates (retained on a filter) and snow water (extracted on a sorbent) might be strongly biased for compounds with some water solubility, which upon sample thawing will dissolve from the particulates into the melt water. This potential artefact needs to be thoroughly addressed as it might render many of the presented results useless. One of the main findings of the study is the unexpected behavior of highly water soluble NMAHs (p11 129-30). Considering the potential experimental bias, I wonder if this finding really holds.

**Authors' response**

We have addressed this concern in our response to comment no. 23 from reviewer 1, since she/he raised the same concern. Please see above our response to that comment for details. Briefly, we have added additional statements to the text, regarding our well-planned analytical method which aimed to minimize sample processing artifacts, as well as evidence from our target compounds dissolution patterns compared to their estimated water solubility. We have ruled out post-sampling dissolution artifacts.

**2) Reviewer's comment**

Snow particulate concentrations are given in ng  $L^{-1}$ , which will strongly depend of the final volume during sample extraction. As this final volume is an arbitrary choice of the authors (and not even reported), it is not clear to me how these concentrations can be used in any reasonable way beyond comparing between different samples. Even the comparison with snow aqueous phase concentrations seems difficult to me. Adding them up to a total snow concentration, as done in Fig. 1 and Table S4A seems hard to justify as well to me.

**Authors' response**

This is a misunderstanding perhaps arising from not reporting the meltwater volumes. The analyte concentrations in each snow phase (aqueous or particulate) were calculated using the volume of melted snow which we measured carefully. Hence, the concentrations in the particulate and aqueous phases can be compared, and can be added together to report total analyte concentrations in snow.

For clarification, we have now added the volumes for melted snow samples to Table 1, and made a note under Section 2.2 (sample processing).

Page 5, Line 20-22: "The volume of meltwater for each sample was determined during filtration using the graduated filter funnel (Fig. S1), and used for calculating the final analyte concentrations in aqueous and particulate phases."

Page 20, Table 1, last column ('snowmelt volume').

**3) Reviewer's comment**

To make my confusion complete, in Table S4B the authors calculate dimensionless total scavenging ratios from the sum of snow particulate and aqueous phase (in ng  $L^{-1}$ ) and the sum of particle and gas phase concentrations (in ng m-3), obtained just before the snow events. This ratio seems to be derived by multiplying ng  $L^{-1}$  concentrations by a factor of 1000 and then dividing by ng m-3 concentrations, ignoring that the denominator in the unit is referring to solvent volume in one case and air volume in the other. This seems very odd to me and needs explanation.

**Authors' response**

The method that we applied for calculating scavenging ratios is a standard method that is well documented in the literature. The multiplication of analyte concentration in snowmelt (ng L-1) by a factor of 1000 is done in order to have the same units in the numerator (ng L-1  $\times$  1000) and denominator (ng m-3), as scavenging ratio is a dimensionless value. To support our approach, we have now cited previous papers that used this method.

Page 4, L7-8: "The efficiency of SVOC wet scavenging is defined by  $W_T$  (unitless) (Bidleman, 1988; Poster and Baker, 1996; Škrdlíková et al., 2011; Shahpoury et al., 2015)"

**4) Reviewer's comment**

The conclusion that the phase partitioning of NMAHs is determined by an interplay between GPP, particle mass size distribution, and dissolution during in- or below-cloud scavenging (abstract 117-19 and p12 113-19) is weak and not convincing. Even assuming the applied methodology was artefact-free and the calculation of the total scavenging ratio can be justified (see above), I cannot follow the authors' reasoning why the observations would indicate an importance of particle mass size distribution and dissolution. This needs to be much better laid out in the discussion and all data in support need to be shown (p12 114-15). Much of this conclusion seems to originate from one single sample, where additional measurements have been made. The poor robustness of results that would follow from this needs to be addressed as well.

**Authors' response**

We agree that results from a single precipitation event cannot be used to draw a strong conclusion; however, the results that the reviewer is referring to are *indication* of an atmospheric process that should not be overlooked. Hence, we have revised the entire paragraph and the related statement in the abstract:

Page12, L8-23: "For the urban site OS, the analyte concentrations in the gas and particulate phases of the air have been determined (sample collected over 24h preceding snowfall onset; Table S4B) in addition to concentrations in precipitation. The scavenging ratios  $W_T$  calculated for the target N/OPAHs and NMAHs (see Eq. 3) were  $10^3 \cdot 10^4$  and  $10^3 \cdot 10^5$ , respectively (Table S4B), which fall within the range suggested for removal of polyaromatic compounds through wet particle scavenging (Shahpoury et al., 2015). With the exception of 1,4- $O_2NAP$  and 9-OFLN, the range of calculated  $W_T$  is consistent with that of modelled  $\Theta$  at 273 K (Table S4B), meaning that the particle scavenging was the dominant removal mechanism. The difference in  $W_T$  between 3- and 5-NSA (~1.4 times higher for 5-NSA) and between 9,10- $O_2ANT$  and OBAT (~1.1 times higher for OBAT) closely resembled that of estimated  $\Theta$  at 273 K (Table S4B). However, we found differences in  $W_T$  between the NMAH subgroups, namely  $W_T$  values were higher for nitrophenols ( $1.3 \times 10^4 - 1.6 \times 10^5$ ) and nitrosalicylic acids ( $5.7 \times 10^4$ –  $8.2 \times 10^4$ ) than nitrocatechols ( $1.1 \times 10^3 - 2.8 \times 10^3$ ; Table S4B), which cannot be explained by the substances' GPP alone (compare  $\Theta$  in Fig. 6C). Although based on a single precipitation event, these results are indication of additional atmospheric processes that NMAHs could undergo, and which may affect the substance wet scavenging. For instance, PM size-dependent mass distribution is a parameter which was suggested to influence the snow scavenging efficiencies (Zhang et al., 2013) – i.e. lower efficiency in the PM size range 0.01-1 µm than for coarse PM ( $\geq 1 \mu m$ ). This parameter should to be taken into account in future studies, and more precipitation episodes need to be considered in order to draw a full picture."

Page 1, L18-19: "*i.e.* NMAHs, is determined by an interplay between gas-particle partitioning in the aerosol and dissolution during in- or below-cloud scavenging."

**5) Reviewer's comment**

On p7 113ff and p12 l6ff the authors discuss acid dissociation in particles in relation to pH. However, they seem to not be aware of sample pH being a different thing than particle pH, i.e. pH of particle liquid water. With a melted snow sample being highly diluted in comparison to aerosol particles, the measured sample pH cannot be used to describe acid dissociation in particle liquid water. This needs to be corrected appropriately.

**Authors' response**

We agree with the reviewer's point of view. We have revised the entire paragraph and removed the statement that the reviewer is referring to. However, under Section 2.6, we do mention the general effect that pKa and pH could have on gas-particle partitioning predictions with our model:

Page 8, L13-22: "One must note that the ppLFER model used here predicts  $K_P$  of a substance in neutral form. In particulate phase, depending on pH of the PM aqueous phase and  $pK_a$  of the target substance, NMAHs may partly or completely deprotonate, resulting in enhanced solubility of the substance in the aqueous phase (Ahrens et al., 2012). This implies that the actual partitioning could be under-predicted for such substances. The effect is expected to be relevant for 5-nitrosalicylic acid (5-NSA; see Table 2 for compound abbreviations),  $pK_a$ : 1.95 at 298 K (Aydin et al., 1997), 3-nitrosalicylic acid (3-NSA; we expect similar  $pK_a$  as that of 5-NSA), 2,4-DNP,  $pK_a$ : 4.07 at 298 K (Lide, 2010), and DNOC,  $pK_a$ : 4.48 at 293 K (WHO, 2000). The rest of NMAHs have noticeably higher  $pK_a$ values – 4-NP: 7.15, 2-methyl-4-nitrophenol (2-M-4-NP): 7.33, 3-methyl-4-nitrophenol (3-M-4-NP): 7.33, 4nitrocatechol (4-NC): 6.93 at 298 K; we expect  $pK_a$  values for 4-methyl-5-nitrocatechol (4-M-5-NC) and 3-methyl-5-nitrocatechol (3-M-5-NC) to be close to that for 4-NC."

**6) Reviewer's comment**

A "dissolved" phase is not a thing. There are particulate, aqueous, or gas phases, for example, but a dissolved phase does not exist and the term needs to be replaced throughout all the manuscript.

**Authors' response**

We have changed the phrase "dissolved" to "aqueous" throughout the text:

Page 1, L11, L16; Page 3, L29, L31; Page 4, L19, L25; Page 5, L14, L15, L22; Page 6, L4, L5; Page 8 L4, L6, L14, L15; Page 9, L3, L4, L5, L7, L11, L12, L16, L34; Page 10, L4, L5, L8, L21; Page 11, L23, L27, L31, L34; Page 12, L5, L7, L30.

**7) Reviewer's comment**

The experimental section lacks many details, e.g. flow rate during extraction (p4 123), volume of ethyl acetate (p5 18-9), volume of buffer (p5, 132-33), concentration of EDTA (p5, 133), duration of ultrasonication (p6 11), volume of buffer (p6 11-2), etc. Please make sure any reader would be able to fully repeat your experiments with the information given.

**Authors' response**

We have added the details mentioned in the reviewer's comment.

Page 5, L19: "A steady sample flow (10 mL min-1) was established..."

Page 6, L6: "...the solvent was exchanged to ethyl acetate  $(3 \times 5 \text{ mL})$ ."

Page 6, L30-31: "The SPE extracts were further concentrated to near dryness using TurboVap II and later dissolved in 1 mL mixture of methanol and 7.15 mM ammonium formate buffer pH 3 (3:7, v/v) containing 400  $\mu$ M EDTA."

Page 6, L33 – Page 7, L1: "Briefly, the particles were extracted using methanol containing 3.4  $\mu$ M EDTA with agitation (3×5 min) in an ultrasonic bath. The final extracts were concentrated to near dryness, and dissolved in 1 mL mixture of methanol and 7.15 mM ammonium formate buffer pH 3 (3/7, v/v), containing 400  $\mu$ M EDTA."

**8) Reviewer's comment**

Data presentation could be improved. All the abbreviations for the different sites are impossible to remember. I suggest removing them completely from the discussion text and referring to the name of the sites instead.

**Authors' response**

It is not possible for us to use the actual names due to their length because they limit the visibility of data in the bar charts. In order to address the reviewer's concern, we have replaced the original abbreviations, which represented the site types, with those that represent the actual site names. These changes have been highlighted throughout the text:

Page 4, L29-L33: "Snow samples were collected between winter 2015 and 2017 from three locations in Germany, i.e. Mainz (MZ15 and MZ17; urban-residential,  $\approx$ 200000 inhabitants), Winterberg (WB) and Altenberg (AB; rural, >10 km from small towns), two locations in Inn Valley, Austria, i.e. Götzens (GS; urban-residential of a mid-sized city, Innsbruck,  $\approx$ 140000 inhabitants) and Kolsassberg (KB; rural, 10-20 km from city and towns), two locations in the Czech Republic, i.e. Ostrava (OS; urban, conurbation with  $\approx$ 700000 inhabitants) and Pustá Polom (PP1 and PP2; rural, 20..."

Page 5, L1-L2: "…km upwind from Ostrava), and one location in the Arctic, Tempelfjorden, Svalbard (TF, remote, 50-80 km from small towns). The sample site details are shown in Table 1. Fresh snow samples (all sites, except TF) were collected by..."

L7: "...which had fallen 3-2 days before, was collected at TF and stored in..."

Page 8, L27: "...for sample site in Svalbard, TF, as the snow fell 2-3 days prior..."

Page 9, L4: "...were found in nearly all samples, except at the remote site TF..." L11: "Overall, TF and MZ17 were the least and most polluted sites..." L19: "... (found in GS, WB, AB and KB) showed the highest concentrations..." L20: "...9-OFLN (found in all but TF;  $\leq 30.3$  ng  $L^{-1}$ )..." L27: "Overall, the remote site TF and rural site KB were the least and most..." L29: "...Snow samples OS, PP1 and PP2 were not phase-separated..."

L33: "...overall higher at the rural sites PP1 and PP2 than the urban site OS (Table S4A)."

L34: "... (i.e. aqueous + particulate) across all sites, TF and PP2 were the least..."

Page 10, L7: "...found in urban samples OS and MZ17 (Fig. 4A) ..." L8: "...PP1, PP2, and TF (Fig. 5A). Overall in the aqueous phase, TF and MZ17 were the least and most..." L11: "...concentrations up to 106.9 ng L-1 (MZ17), followed..." L12: "...the only NMAH found at the remote site TF." L14: "...urban site MZ17 (Fig. 4B). Overall in the particulate phase, the remote site TF was the cleanest..." L15: "...the urban site MZ17 was the most polluted..."

L24: "...higher than that measured for the remote sample TF"

L25: "In urban samples with the exception of OS..."

L28: "...KB and PP1, and the remote sample TF, nitrosalicylic acids..."

Page 11, L1: "...values for MZ17 and OS sampling events were..." L16: "The only exception was 9,10-O2ANT in MZ15 and MZ17 samples." L29: "...this was more pronounced in urban and rural samples OS, PP1 and PP2."

Page 12, L8: "For the urban site OS, the analyte concentrations..."

Page 20, Table 1, column 1 Page 22-28, Figure 1-7.

**- Reviewer's comment**

Other abbreviations are used without explanation, e.g. the indices WSOM and OP (p6, 129).

**Authors' response**

The abbreviation WSOM and OP were actually defined in the text:

Page 7, L19-20: "... for absorption into water soluble organic matter (WSOM) and organic polymers (OP)..."

**- Reviewer's comment**

Different panels in the Figures are sometimes difficult to compare, because the order of compounds changes.

**Authors' response**

We have remade all the graphs and ordered the substance names in a consistent way:

Page 22-28, Figure 1-7.

**- Reviewer's comment**

Figure 3 contains empty brackets in the y-axis label, Figure 4 even contains only empty brackets as y-axis label.

**Authors' response**

Empty brackets indicate dimensionless parameters. Since we have remade all figures, Figure 3 and 4 are now referred to as Figure 6 and 7. We have revised y-axis labels in these figures:

Page 27, Figure 6 Page 28, Figure 7

**- Reviewer's comment**

The discussion of air mass back trajectories in 3.1 is odd, as it is nowhere else in the manuscript referred to. If still important, it should be substantiated.

**Authors' response**

We have moved the discussion of air mass history analysis to the Supplement (Page S5, Text S1), and made a reference in the text:

Page 9, L10-11: "Potential source areas for such emissions are reflected in the air mass histories of all samples (see Text S1 and Fig. S2)."

**9) Reviewer's comment**

Correction factors for fOM (p7 11-2) are taken from a study in the 1980s done in Los Angeles, USA. I wonder if this is really the best reference for the sites sampled by the authors. Also, these factors are likely to differ substantially both between sites but also between different meteorological situations. A note on the sensitivity of the results on such highly uncertain parameters would be helpful.

**Authors' response**

Studies reporting comprehensive chemical composition of aerosol organic matter are extremely rare and the reference we used, Rogge et al., 1993, is amongst a few available in the literature, which reflect empirical data across sites. In a previous publication, we conducted a sensitivity study regarding the allocation of  $f_{OM}$ , and compared our assumptions with measured  $f_{OM}$  sub-fractions from a central European site. We have now added a statement in the text and cited the publication containing the sensitivity analysis:

Page 8, L1-2: "These factors were previously verified through a sensitivity study (Shahpoury et al., 2016)."

**10) Reviewer's comment**

Sections 3.2 and 3.3 are tedious to read with all the abbreviations and very detailed concentrations. I suggest discussing the main observations in these measurements in a more compact way and leaving much of the numbers to the corresponding Figures and Tables. Also, some paragraphs might be moved to the Introduction (e.g. p8 114-22). The space gained here should be used to discuss main findings of the paper in a more substantiated way (see comment above).

**Authors' response**

We agree with the reviewer and, as suggested, "p8 114-22" has been moved to Introduction section. The same has been done for NMAHs:

Page 2, L17-29: "For instance, 9-fluorenone (9-OFLN), 9,10-anthraquinone (9,10-O2ANT), 1,4-naphthoquinone (1,4-O2NAP), and 1,2-benzanthraquinone (1,2-O2BAA) were previously found in diesel exhaust (Choudhury, 1982; Cho et al., 2004)..... formation of 1,4-O2NAP and 9,10-O2ANT following photolysis of 1-NNAP and 9-NANT was suggested by previous studies (Keyte et al., 2013; Bandowe and Meusel, 2017)."

Page 3, L6-11: "Increased production of nitrocatechols in the urban environment due to ......biomass burning as the major emission source, similar to the previous reports on nitrocatechols (Iinuma et al., 2010; Kitanovski et al., 2012; Kahnt et al., 2013; Chow et al., 2016; Caumo et al., 2016)."

In addition, we removed the discussion of less significant results and highlighted our major findings under Section 3.1 and 3.2:

[revised manuscript text omitted]
_{\rm BC}$ ,  $a_{\rm (NH_4)_2SO_4}$  and  $a_{\rm NaCl}$  are the adsorbent specific surface areas (m2surface g-1adsorbent), and  $f_{\rm BC}$ ,  $f_{(NH_d)$ -SO4 and  $f_{NaCl}$  are their mass mixing ratios in PM (gadsorbent g-1PM). For  $a_{BC}$ , the geometric mean of 18.21 m2 g-1 was calculated from the values reported for traffic, wood, coal, and diesel soot (Jonker and Koelmans, 2002), whereas,  $a_{(NH_4)2SO_4}$  and  $a_{NaCl}$  of 0.13 and 0.10 m2 g-1 were taken from Goss et al., (2003).  $K_{DMSO}$  (m3air m-3DMSO) and  $K_{PU}$  (m3air  $g^{-1}_{PU}$ ) are the substance partitioning (absorption) coefficients for dimethyl sulfoxide-air and polyurethane-air partitioning systems;  $\rho_{\text{DMSO}}$  is dimethyl sulfoxide density (g m-3);  $f_{\text{WSOM}}$  and  $f_{\text{OP}}$ , are mass mixing ratios of absorbing phases ( $g_{absorbent}$  g-1PM), corresponding to  $f_{OM}$  (the mixing ratio of total organic matter in PM) × 0.60 and  $f_{OM}$  × 0.40, respectively. The correction factors of 0.60 and 0.40 were estimated based on the data from Rogge et al., (1993). These factors were previously verified through a sensitivity study (Shahpoury et al., 2016). We assumed two scenarios for model calculations:  $f_{BC} = 0.03$  and  $f_{OM} = 0.30$ , and  $f_{BC} = 0.06$  and  $f_{OM} = 0.60$ . This resulted in  $f_{WSOM}$  and  $f_{OP}$  of 0.18 and 0.12, and 0.36 and 0.24 for the two scenarios, respectively. The contributions of inorganic salts and PM aqueous

- 5 phase to the overall sorption process were neglected; we acknowledge that under high relative humidity, salts may be present in aqueous phase, and subject to liquid-liquid phase separation with PM organic matter (You et al., 2014). The individual partitioning constants used in the multi-phase model were calculated using substance-specific Abraham descriptors listed in Table S2 and ppLFER models listed in Table S3 (Abraham et al., 2010; Kamprad and Goss, 2007; Roth et al., 2005; Goss et al., 2003).
- 10 See Shahpoury et al., (2016) for more details about calculation with multiphase model and Endo and Goss (2014) for background about ppLFER concept. The predicted  $K_P$  values were converted to  $\Theta$  under two scenarios with  $c_{PM}$  of 25 and 50 µg m-3, Eq. (5):

$$\theta = \frac{K_{\rm P} \, c_{\rm PM}}{(1 + K_{\rm P} \, c_{\rm PM})} \tag{5}$$

One must note that the ppLFER model used here predicts  $K_P$  of a substance in neutral form. In particulate phase, depending on pH of the PM aqueous phase and  $pK_a$  of the target substance, NMAHs may partly or completely

- 15 deprotonate, resulting in enhanced solubility of the substance in the aqueous phase (Ahrens et al., 2012). This implies that the actual partitioning could be under-predicted for such substances. The effect is expected to be relevant for 5nitrosalicylic acid (5-NSA; see Table 2 for compound abbreviations),  $pK_a$ : 1.95 at 298 K (Aydin et al., 1997), 3nitrosalicylic 
[revised manuscript text omitted]
                                    |                      |               |                     |                             |                        |
| <mark>MZ15</mark> , Mainz                | 49.99° N, 8.23° E    | 23 Feb 2015   | 8:00                | 12:45                       | <mark>0.4</mark>       |
| <mark>GS</mark> , Götzens                | 47.23° N, 11.31° E   | 25 Feb 2015   | Overnight           | 9:00                        | <mark>0.45</mark>      |
| <mark>OS</mark> , Ostrava                | 49.86° N, 18.26° E   | 19 Feb 2016   | 14:00               | 19:00                       | <mark>0.5</mark>       |
| <mark>MZ17</mark> , Mainz                | 49.99° N, 8.23° E    | 10 Jan 2017   | 9:00                | 15:00                       | 1                      |
| Rural                                    |                      |               |                     |                             |                        |
| WB, Winterberg                           | 51.18° N, 8.49° E    | 03 Mar 2015   | 15:00               | 18:30                       | <mark>0.5</mark>       |
| AB, Altenberg                            | 50.78° N, 13.69° E   | 05 Mar 2015   | Overnight           | 8:00                        | <mark>0.5</mark>       |
| KB, Kolsassberg                          | 47.28° N, 11.65° E   | 25 Feb 2015   | Overnight           | 10:00                       | <mark>0.5</mark>       |
| PP1, Pustá Polom 1                       | 49.86° N, 17.98° E   | 19 Feb 2016   | 9:30                | 23:00                       | <mark>0.5</mark>       |
| <mark>PP2</mark> , Pustá Polom 2         | 49.86° N, 17.98° E   | 23 Feb 2016   | 14:00               | 23:00                       | <mark>0.5</mark>       |
| Remote                                   |                      |               |                     |                             |                        |
| TF, Tempelfjorden                        | 78.45° N 17.32° E    | 4 Mar 2015    | 1 Mar 2015          | after snowfall a | <mark>1.65</mark>      |
| a old snow, fallen 3-2 | days before sampling |               |                     |                             |                        |

**Table 1. Sampling site details**

| Analyte                  | Abbreviation            | RT    | Q1    | Detection a |
|--------------------------|-------------------------|-------|-------|-------------------------------|
| 1-Nitronaphthalene       | 1-NNAP                  | 12.12 | 173.1 | P (86)                        |
| 2-Nitronaphthalene       | 2-NNAP                  | 12.62 | 173.1 | P (43)                        |
| 5-Nitroacenaphthene      | 5-NACE                  | 17.52 | 199.1 | n.d.                          |
| 2-Nitrofluorene          | 2-NFLN                  | 19.07 | 211.1 | <mark>n.d.</mark>             |
| 9-Nitroanthracene        | 9-NANT                  | 19.46 | 223.1 | P (71)                        |
| 9-Nitrophenanthrene      | 9-NPHE                  | 20.64 | 223.1 | n.d.                          |
| 3-Nitrophenanthrene      | 3-NPHE                  | 21.4  | 223.1 | <mark>n.d.</mark>             |
| 2-Nitrofluoranthene      | 2-NFLT                  | 25.75 | 247.1 | <mark>P (71)</mark>           |
| 3-Nitrofluoranthene      | 3-NFLT                  | 25.80 | 247.1 | <mark>n.d.</mark>             |
| 1-Nitropyrene            | 1-NPYR                  | 26.63 | 247.1 | <mark>n.d.</mark>             |
| 2-Nitropyrene            | 2-NPYR                  | 26.95 | 247.1 | <mark>n.d.</mark>             |
| 7-Nitrobenz(a)anthracene | 7-NBAA                  | 29.41 | 273.1 | <mark>n.d.</mark>             |
| 6-Nitrochrysene          | 6-NCHR                  | 30.66 | 273.1 | <mark>n.d.</mark>             |
| 1,3-Dinitropyrene        | $1,3-N_2PYR$            | 31.8  | 292.1 | <mark>n.d.</mark>             |
| 1,6-Dinitropyrene        | $1,6-N_2PYR$            | 32.81 | 292.1 | <mark>n.d.</mark>             |
| 1,8-Dinitropyrene        | $1,8-N_2PYR$            | 33.54 | 292.1 | <mark>n.d.</mark>             |
| 6-Nitrobenz(a)pyrene     | 6-NBAP                  | 36.73 | 297.1 | <mark>n.d.</mark>             |
| 1,4-Naphthoquinone       | 1,4-O 2 NAP  | 10.18 | 158.1 | <mark>A (14)</mark>           |
| 9-Fluorenone             | 9-OFLN                  | 13.78 | 180.1 | <mark>A (100) P (86)</mark>   |
| 9,10-Anthraquinone       | 9,10-O 2 ANT | 17.03 | 208.1 | <mark>A (86) P (57)</mark>    |
| Acenaphthenequinone      | O 2 ACE      | 17.82 | 198.1 | <mark>A (100) P (100)</mark>  |
| 2-Nitro-9-fluorenone     | 2-N-9-OFLN              | 20.54 | 225.1 | n.d.                          |
| Benz(a)fluorenone        | BaOFLN                  | 22.88 | 230.1 | <mark>A (29) P (100)</mark>   |
| Benz(b)fluorenone        | BbOFLN                  | 23.82 | 230.1 | <mark>Р (86)</mark>           |
| Benzanthrone             | OBAT                    | 25.07 | 230.1 | <mark>A (29) B (43)</mark>    |
| 1,2-Benzanthraquinone    | $1,2-O_2BAA$            | 26.46 | 258.1 | <mark>A (71) B (100)</mark>   |
| 3-Nitrosalicylic acid    | 3-NSA                   | 3.60  | 182   | <mark>A (100) P (11)</mark>   |
| 5-Nitrosalicylic acid    | 5-NSA                   | 5.07  | 182   | <mark>A (100) P (11)</mark>   |
| 4-Nitrocatechol          | 4-NC                    | 7.76  | 154   | <mark>A (90) P (33)</mark>    |
| 4-nitroguaiacol          | 4-NG                    | 8.29  | 168   | <mark>A (50) P (22)</mark>    |
| 4-Methyl-5-nitrocatechol | 4-M-5-NC                | 9.47  | 168   | A (90) P (22)                 |
| 4-Nitrophenol            | 4-NP                    | 10.00 | 138   | <mark>A (100) P (100)</mark>  |
| 2,4-Dinitrophenol        | 2,4-DNP                 | 10.92 | 183   | <mark>A (100) P (33)</mark>   |
| 3-Methyl-4-nitrophenol   | 3-M-4-NP                | 13.19 | 152   | A (100) P (89)                |
| 3-Methyl-5-nitrocatechol | 3-M-5-NC                | 14.01 | 168   | A (100) P (22)                |
| 2-Methyl-4-nitrophenol   | 2-M-4-NP                | 16.72 | 152   | <mark>A (100) P (89)</mark>   |
| Dinitro-ortho-cresol     | DNOC                    | 17.05 | 197   | A (100) P (78)                |

Table 2. Target compound list

Abbreviations, retention times (RT), and quantification ions (Q1) of target analytes; *a* analyte detection in aqueous (A) or particulate (P) phase; numbers in brackets indicate percentage of detection across the samples in each phase; n.d.: not detected